# Investigation of occupational risk factors for the development of non-Hodgkin's lymphoma in adults: A hospital-based case-control study

**Marcia Sarpa**[1,2]*, **Vanessa Índio do Brasil da Costa**[1], **Sâmila Natiane Ferreira**[1,2], **Carolina Ávila de Almeida**[1,2], **Paula Gabriela Sousa de Oliveira**[1,2], **Letícia Vargas de Mesquita**[1,2], **Arthur O. C. Schilithz**[1], **Claudio Gustavo Stefanoff**[3], **Rocio Hassan**[4‡], **Ubirani Barros Otero**[1]

**1** Coordination of Prevention and Surveillance, Brazilian National Cancer Institute, Rio de Janeiro, Brazil, **2** Environmental Mutagenesis Laboratory, Federal University of Rio de Janeiro State (UNIRIO), Rio de Janeiro, Brazil, **3** Coordination of Clinical Research and Technological Incorporation, Brazilian National Cancer Institute, Rio de Janeiro, RJ, Brazil, **4** Oncovirology Laboratory, Bone Marrow Transplantation Center, Brazilian National Cancer Institute (INCA), Rio de Janeiro, Brazil

‡ In memorium.
* marcia.sarpa@inca.gov.br, marciasarpa@gmail.com

**Data Availability Statement:** All relevant data are within the paper.

## Abstract

Non-Hodgkin's Lymphoma (NHL) is a malignancy of the lymphoid lineage of the hematopoietic system has worldwide, especially in developed countries. Better diagnostic and recording techniques, longer life expectancy, and greater exposure to risk factors are hypotheses for this growing incidence curve. Occupational exposures to chemical, biological, and physical agents have also been associated with NHL development, but the results are still controversial. We have investigated the occupational and lifestyle case-control study design with 214 adult patients and 452 population controls. Socio-demographic, clinical, and occupational exposure data were obtained through individual interviews with a standardized questionnaire. Clinical, laboratory, and histopathological data were obtained through medical records. Risk of NHL (any subtype), B-cell lymphoma, DLBCL, Follicular lymphoma and T-cell lymphoma was elevated among the those who had ever been exposed to ***any solvents***, ***hydrocarbon solvents***, ***pesticides***, ***meat and meat products***, and ***sunlight*** and tended to increase by years of exposure. A significant upward trend with years of exposure was detected for ***any solvents*** and ***hydrocarbon solvents*** (NHL (any subtype) p-value for trend<0.001), B-cell lymphoma (p-value for trend<0.001), and T-cell lymphoma (p-value for trend<0.023), ***pesticides*** (NHL (any subtype), p for trend<0.001) and T-cell lymphoma (p for trend<0.002), ***meat and meat products*** (NHL (any subtype) (p for trend<0.001) and DLBCL (p for trend<0.001), and ***sunlight*** (B-cell lymphoma (p for trend<0.001). The results of this study agree line with other international studies, can be extrapolated to other countries that have the same socio-demographic and occupational characteristics as Brazil and support strategies for surveillance and control of work-related cancer.

**Funding:** Vanessa Indio do Brasil da Costa received an institutional development scholarship from Brazilian National Cancer Institute (INCA); Sâmila Natiane Ferreira and Letícia Vargas de Mesquita received a scholarship from CNPQ/INCA; Carolina Ávila de Almeida and Paula Gabriela Sousa de Oliveira received a scholarship from FAPERJ; The Pan-American Health Organization (PAHO) financed the study through the INCA/PAHO TC 43 agreement. The funders had no role in study design, data collection and analysis, decision to publish, or preparation of the manuscript.

**Competing interests:** The authors have declared that no competing interests exist.

## Introduction

Non-Hodgkin's lymphomas (NHLs) are a group of malignant diseases with distinct genetic, morphological, and clinical characteristics [1]. They originate from progenitor cells and mature B cells, from progenitor and mature T-cells or from natural killer cells (rarely), and vary in pathophysiology, histological appearance, clinical course, and response to therapy [2]. The most common subtypes are diffuse large B-cell and follicular lymphomas, together accounting which account for approximately 30% and 20% of all NHL cases, respectively [3, 4]. Other less common subtypes are include Burkitt lymphoma (1–5% of adult lymphoma cases); lymphoplasmacytic lymphoma (rare and indolent); small cell lymphocytic lymphoma; marginal zone lymphoma (5%–10% of NHL cases); mantle cell lymphoma (2%–4% of NHL cases); peripheral T-cell lymphoma, which the WHO defines as "natural killer", mature T-cell neoplasms, and Mycoses fungoides/Sezary syndrome [1].

The highest incidence rates of NHL are found in industrialized developed countries in North America and Europe, in addition to Australia and New Zealand ($> 7.0/100,000$ thousand people, in for both men and women) [5]. Each year, an estimated 509,590 new cases of NHL (2.7% of total cancer) and around 250,000 deaths (2.4% of total deaths) occur worldwide. NHL is among the top ten most common types of cancer in Brazil, each per year [6]. For the triennium 2023–2025 in Brazil, it is estimated that 6,420 new cases will occur in men and 5,620 new cases will occur in women annually in Brazilian men and women, respectively per year between 2023 and 2025. This corresponds to an estimated rate of 6.08 new cases per 100,000 men and 5.08 per 100,000 women. The highest incidence rates of NHL are observed in men in the Southeast region (8.2/ per 100,000 thousand) and in women in the South Region (7.3/ per 100,000 thousand) [7].

The best-established risk factors for NHL are immunosuppression; infections, including HIV/AIDS virus, hepatitis C, Epstein- Barr; and autoimmune diseases [8]. The proportion of NHL cases related to celiac disease, systemic lupus erythematosus (SLE), Sjogren's syndrome, and rheumatoid arthritis (RA), for example, is considered small, due to the low prevalence of these diseases in the general population [4, 9, 10]. Hereditary factors are also cited, but they are responsible for only a small portion of NHL cases [10]. Other factors for the development of NHL include smoking (follicular lymphoma) and occupation, especially occupational exposure to specific chemical, physical, and biological agents [11].

The chemical agents most consistently associated with NHL, mentioned in the literature, are solvents and pesticides [8, 12, 13]. Other occupational factors related to NHL include sunlight [14]; ionizing radiation [13]; and organic dust (grain, leather, fabric, and wood dusts) [15, 16]; However, evidence of the association of these factors with NHL is inconsistent [8].

The International Agency for Research on Cancer (IARC) classifies 87 occupational exposures in groups 1 and 2A [14] (68 from group 1 and 19 from group 2A). Nine of these exposures were specifically associated with non-Hodgkin's lymphoma.

Thus, the aim of this study was to describe the socio-demographic and clinical data of patients treated at the Brazilian National Cancer Institute (INCA), in addition to assess whether there is an association between occupational exposures to chemical, physical and biological agents and the chance of developing NHL in these patients. Also, it is expected that the results of this study will support the implementation of actions to prevent and control exposure to carcinogens in the work environment.

## Materials and methods

### Study design and population

This is a hospital-based case-control study, with patients residing in the state of Rio de Janeiro (RJ, Brazil).

It is noteworthy that 17.5 million people live in the across its 92 municipalities in the state of Rio de Janeiro (RJ, Brazil). Therefore, although the fact that the case-control study was carried conducted out in a single institute in the country, this institute is responsible for providing care to around 7.000 new patients per year and is also responsible for formulating public policies on cancer in Brazil.

The sample size calculation was performed using the EpiInfo software suite, version 3.3.2, using the calculation basis for case-control studies, with a confidence interval of 95%. Considering human exposure to the main carcinogens described in the literature associated with lymphomas, the sample size was determined using with the following parameters: $\alpha = 0.05$; 2:1 case-control ratio; and a 5% prevalence of occupational hazards among controls (adopting a more conservative measure; OR = 2.5). Thus, 215 cases and 430 controls would be required to achieve a statistical power of 80%.

Inclusion criteria for the cases were included: age between 25 and 75 years; being a resident of the state of Rio de Janeiro; having a Brazilian nationality; with a primary diagnosis of NHL confirmed by laboratory and histopathological tests. Exclusion criteria for the cases were included: patients diagnosed with HIV were not included.

Inclusion criteria for the control group were included: visitors or companions of patients at Cancer Hospital I (HCI/INCA), aged between 25 and 75 years, without previous or current diagnosis of any type of cancer and not accompanying or visiting patients diagnosed with hematological cancer.

During the study, 274 cases and 557 controls matched to cases by gender and age-group (± 10 years) were invited to participate in the study. Of these, 10 cases (3.6%) refused to participate in the study and 50 cases (18.2%) did not meet the previously established eligibility criteria. Regarding controls, 105 (18.9%) did not meet the eligibility criteria or refused to participate in the study. As most visitors of patients diagnosed with non-haematological neoplasms at the hospital's outpatient clinics or wards were female, it was necessary to recruit male blood donors (N = 143) at the hospital Haemotherapy Service. A total of 214 NHL cases (112 men and 102 women) and 452 controls (207 men and 245 women) were included in the study. The complete diagram with the initial and final number of study participants, including refusals, can be seen in Fig 1.

Therefore, this study includes 214 incident patients between the ages of 25 and 75 years with NHL (International Classification of Disease—ICD10; C82-85) diagnosed between 2013 and 2016. Hospital-based cancer registries of cases were identified, and histopathological reports were examined to confirm the diagnosis of NHL, based on the WHO classification: Follicular non-Hodgkin's lymphoma (C82) N cases; Diffuse non-Hodgkin's lymphoma (C83) N cases; Cutaneous and peripheral T-cell lymphomas (C84) N cases; non-Hodgkin's lymphoma of other and unspecified types (C85) N cases [17].

## Data collection

The interviews of the research participants were carried out by qualified and trained professionals from the Technical Area of Environment, Work and Cancer of the National Cancer Institute (INCA; Ministry of Health; Brazil). On the day of the medical appointment or some routine procedure at the hospital, the patients were invited to participate in the research by professionals from the INCA research group. Those patients who agreed to participate in the research signed the Free and Informed Consent Form (TCLE). Data collection was carried out through a standardized and pre-tested questionnaire with questions aimed at obtaining socio-demographic, clinical, and occupational information.

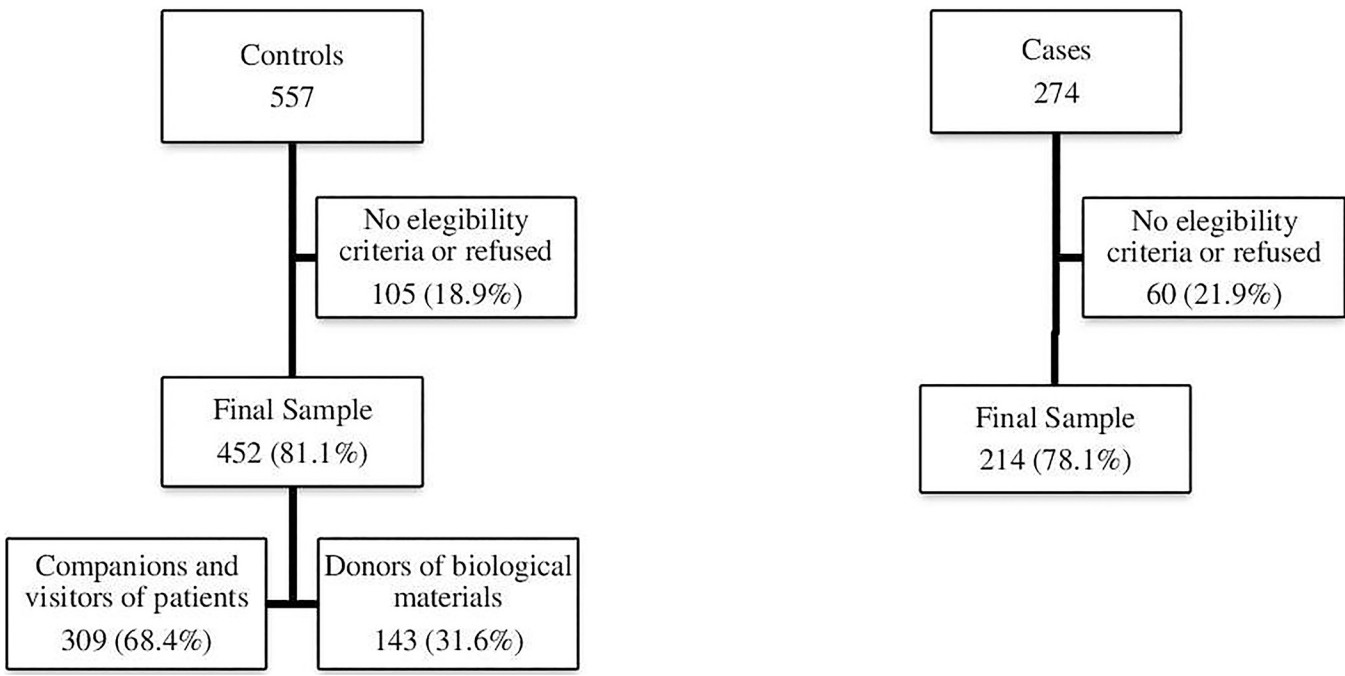

**Fig 1. Research participants flow diagram.**

Participants filled out an occupational history questionnaire detailing all their activities and exposure to chemical, physical, or biological agents. Questions included: "In what year did you start this job?" "In what year did you leave this job?" "What was your position in this job?" "What economic activity did you perform for the company/industry?" "In this job, were you exposed to benzene (yes/no); if so, how often (daily, weekly, rarely, no contact). Regarding solar radiation, the main question was: "Do you or did you work in sunny or open-air environments?"

In the same period, companions and visitors of patients diagnosed with non-hematological neoplasms undergoing medical care at the outpatient clinics or wards of the Cancer Hospital I (HCI) of the National Cancer Institute (INCA) were invited to participate in the research to compose the control group. In addition, there was a need to recruit donors of biological materials in the Hemotherapy Service (HCI/INCA), since most of the companions and visitors of patients diagnosed with non-hematological neoplasms undergoing medical care at the hospital's outpatient clinics or wards HCI/INCA were female. Therefore, on recruitment days, donors of biological materials at the Hemotherapy Service (HC1/INCA) were also invited to participate in the research.

The main outcome of the study was total NHL, and secondary outcomes, such as subtypes B-cell lymphoma subtypes, DLBCL, and T-cell lymphoma were also included in the study. Occupational exposures were considered as explanatory variables. In addition to these exposures, the literature mentions age, gender, race/ethnicity, family history, autoimmune diseases, and some infections as risk factors for NHL.

To characterize the study population and control possible confounding factors, the following information was collected: demographic and clinic variables, including "year of interview" (2013, 2014, 2015 or 2016), "sex" (male, female), "age" (20–39; 40–59; > = 60), "region within the state of Rio de Janeiro" (metropolitan, others), "*years at the current residence*" ($\leq$ 20, >20), "education" (no degree of education or primary school, high school, university degree),

"employment actual": yes (private institution, retired, self-employed, public institution, unemployed, dependent, pensioner, sick leave), "marital status (no companion, with companion, widowed), "family income" (minimum salaries/wages).

Among the lifestyle variables, we considered the following: "smoking habits" (never smoked, current smoker, past smoker: $\leq$ 10 years, past smoker: 10+ years), "smoke load: pack years" (never, daily: <10years, 10+years, others), "alcohol consumption (current)" (non-drinking, 1 to 5 drinks per week, > 5 drinks per week), "cellular phone use" (never, daily: $\leq$ 10 years, daily: 10+years, others).

The occupational history was limited to their main occupations in the last 20 working years and all occupations lasting one year or more were considered. Information on occupational exposure to chemical substances, physical and biological agents possibly associated with the risk of developing NHL such as solvents, paints, metals, pesticides, cleaning products, biological material, meat and meat products, industrial radiation, and sunlight (never, < = 10 years, 10+ years) were self-reported collected. The participant responded about which agents he was exposed to and for how long, in the last 20 years, for each reported occupation. Although the questionnaire contains this information, it was decided to present the results keeping the exposures grouped (any solvents, hydrocarbons solvents, pesticides, metals, meat and meat products, and sunlight), without specifying which solvents, metals or pesticides were reported.

## Statistical analysis

A descriptive analysis was performed to verify the profile of the studied population. The variables were categorized according to the cutoff points pre-established in the literature. To assess differences in the distributions of categorical variables between the comparison groups (cases and controls), Pearson's chi-square tests or Fisher's exact test (for cases where the variable was dichotomous and there were less than five individuals in a cell) were performed. Missing values were excluded from the analysis and the information was reported in Tables 1 and 2. For continuous variables, normality was tested (Kolmogorov test). When the variable had a normal distribution, Student's t-test was used to compare the means; otherwise, the Mann-Whitney non-parametric test was used.

By unconditional logistic regression, multivariate models were estimated to analyze exposure variables to chemical, physical, and biological agents as well as the risk of NHL according to the cases "Total NHL", "B-cell lymphoma", "DLBCL", "Follicular lymphoma", and "T-cell lymphoma". The covariates used in the study were: gender, age, race, family income, place of residence, level of education, smoking, alcohol intake, past illnesses, and family history of hematological cancer, including the history of NHL (described in the "Data collection" section and in the footer of Table 3). The first step was select the covariates to be included in the final regression model, to each outcome and exposure, in the univariate analyses with NHL at p-value < 0.250 [18]. After, to the multivariate analysis, we followed a manual stepwise forward procedure (most significant variables entered first), where the occupational exposure of interest was tested, and the adjustment of each model was evaluated by inserting each of the previously selected covariates.

The Odds Ratio and its 95% confidence interval were calculated as the measure of association; the null hypothesis was rejected if the 95% confidence interval did not include unity. The software used was SPSS 25.0.

The letters of informed consent, questionnaires, and all other correspondence with study participants were approved by the INCA's Research Ethics Committee under registration number 135/11.

**Table 1. Sociodemographic characteristics of the participants in a case-control study, RJ, Brazil, 2013–2016.**

| Characteristic | Controls | | Cases | | p-value |
|---|---|---|---|---|---|
| | (n = 452) | | (n = 214) | | |
| | n | % | n | % | |
| **Year of the interview** | | | | | 0.058 |
| 2013 | 116 | 25.7 | 58 | 27.1 | |
| 2014 | 90 | 19.9 | 56 | 26.2 | |
| 2015 | 163 | 36.1 | 76 | 35.5 | |
| 2016 | 83 | 18.34 | 24 | 11.2 | |
| **Sex** | | | | | 0.135 |
| Male | 207 | 45.8 | 112 | 52.3 | |
| Female | 245 | 54.2 | 102 | 47.7 | |
| **Age** | | | | | 0.063 |
| 20–39 | 85 | 18.8 | 33 | 15.3 | |
| 40–59 | 235 | 52.0 | 99 | 46.1 | |
| $\geq 60$ | 132 | 29.2 | 82 | 38.6 | |
| **Region within the state of Rio de Janeiro** | | | | | **< 0.001** |
| Metropolitan | 426 | 94.2 | 171 | 79.9 | |
| Others | 26 | 5.8 | 43 | 20.1 | |
| **Years of residence (Rio de Janeiro State)** | | | | | 0.671 |
| $\leq 20$ | 267 | 59.5 | 103 | 57.8 | |
| $> 20$ | 182 | 40.5 | 90 | 42.2 | |
| No information | 3 | | 1 | | |
| **Education** | | | | | **< 0.001** |
| No degree of education or primary school | 124 | 27.5 | 101 | 47.1 | |
| High school | 194 | 43.0 | 81 | 37.9 | |
| University degree | 133 | 29.5 | 32 | 15.0 | |
| No information | 1 | | | | |
| **Employment (actual)** | | | | | |
| Private institution (yes) | 111 | 24.5 | 25 | 11.7 | **< 0.001** |
| Retired (yes) | 97 | 21.5 | 61 | 28.6 | **0.042** |
| Self-employed (yes) | 96 | 21.2 | 21 | 9.9 | **0.001** |
| Public Institution (yes) | 71 | 15.7 | 14 | 6.6 | **0.001** |
| Not employed (yes) | 63 | 13.9 | 25 | 11.6 | 0,422 |
| Dependent (yes) | 16 | 3.5 | 11 | 5.1 | 0,322 |
| Pensioner (yes) | 13 | 2.8 | 14 | 6.5 | 0.023 |
| Sick leave (yes) | 7 | 1.5 | 59 | 27.4 | **< 0.001** |
| **Marital status** | | | | | 0.075 |
| No companion | 125 | 27.7 | 73 | 34.2 | |
| With companion | 309 | 68.5 | 129 | 60.2 | |
| Widowed | 17 | 3.8 | 12 | 5.6 | |
| No information | 1 | | 0 | | |
| **Family income (Minimum salary)** | | | | | **< 0.001** |
| Mean (Std. Dev) | | 4.1 (5.9) | | 2.7 (3.6) | |

Values correspond to the absolute frequencies of the variables.

The p-value is related to the comparison of variable proportions between NHL cases and controls, using the chi-square or Fisher test (for strata with absolute number < 5).

The p-value corresponds to the comparison of variable means between NHL cases and controls using the Mann-Whitney test

In bold, values with a statistically significant result (< 0.05). 1 minimum salary in Brazil in 2016 was equal to 880 reais (220 USD).

* Missing values cases = 1

**Table 2. Habits of the participants in a case-control study, RJ, Brazil, 2013–2016.**

| Characteristic | Controls | | Cases | | p-value |
|---|---|---|---|---|---|
| | (n = 452) | | (n = 214) | | |
| | n | % | n | % | |
| **Smoking status** | | | | | 0.074 |
| Never smoked | 283 | 65.5 | 117 | 57.1 | |
| Current smoker | 46 | 10.6 | 20 | 9.8 | |
| Past smoker ($\leq$ 10 years) | 39 | 9.0 | 30 | 14.6 | |
| Past smoker (10+ years) | 64 | 14.9 | 38 | 18.5 | |
| No information | 20 | | 9 | | |
| **Smoke load (Pack years)** | | | | | **< 0.001** |
| Never | 283 | 65.7 | 117 | 57.1 | |
| $\leq$ 20 | 93 | 21.5 | 34 | 16.6 | |
| > 20 | 53 | 12.3 | 54 | 26.3 | |
| No information | 23 | | 9 | | |
| **Alcohol consumption (current)** | | | | | **0.003** |
| Non-drinker | 204 | 46.9 | 120 | 60.6 | |
| 1 to 5 drinks per week | 174 | 40.0 | 53 | 26.8 | |
| > 5 drinks per week | 57 | 13.1 | 25 | 12.6 | |
| No information | 17 | | 16 | | |
| **Cellular use** | | | | | **< 0.001** |
| Never | 22 | 4.9 | 27 | 13.5 | |
| Daily ($\leq$ 10 years) | 144 | 32.1 | 50 | 24.0 | |
| Daily ($10^{+}$years) | 224 | 49.9 | 86 | 41.3 | |
| Others | 59 | 13.3 | 44 | 21.2 | |
| No information | 3 | | 7 | | |
| **Wireless Phone use** | | | | | **< 0.001** |
| Never | 152 | 34.0 | 106 | 52.2 | |
| Daily (< 10 years) | 61 | 13.6 | 25 | 12.2 | |
| Daily ($10^{+}$years) | 94 | 21.1 | 30 | 14.6 | |
| Others | 140 | 31.3 | 43 | 21.0 | |
| No information | 5 | | 10 | | |

Values correspond to the absolute frequencies of the variables.

The p-value is related to the comparison of variable proportions between NHL cases and controls, using the chi-square or Fisher test (for strata with absolute number < 5).

The p-value corresponds to the comparison of variable means between NHL cases and controls using the Mann-Whitney test

In bold, values with a statistically significant result (< 0.05).

*Missing values: controls = 2; cases = 8

## Results

This is a hospital-based case-control study, with patients residing in the state of Rio de Janeiro (RJ, Brazil). Care was provided and were seen at the Brazilian National Cancer Institute (INCA).

Regarding the cases of NHL, the predominant histological type was B-cell lymphoma (185 cases; 86.0%) and T-cell lymphoma were 30 cases (14.0%).

There was no statistical difference in age (p = 0.063) between the two groups. Most individuals were older than 40 years, with mean age of cases (mean age: 56 years; standard deviation,

**Table 3. Occupational risk factors and development of total NHL, B-cell lymphoma, DLBCL, follicular lymphoma and T-cell lymphoma in the participants in a hospital-based case-control study, RJ, Brazil, 2013–2016.**

| Risk Factors | Total NHL | | B-cell lymphoma | | DLBCL | | Follicular lymphoma | | T-cell lymphoma | |
|---|---|---|---|---|---|---|---|---|---|---|
| | Controls/ Cases | OR adjusted 95% CI | Controls/ Cases | OR adjusted 95% CI | Controls/ Cases | OR adjusted 95% CI | Controls/ Cases | OR adjusted 95% CI | Controls/ Cases | OR adjusted 95% CI |
| *Chemical* | | | | | | | | | | |
| **Any solvents** | | | | | | | | | | |
| Never | 275/93 | 1.0 | 275/83 | 1.0 | 275/38 | 1.0 | 275/15 | 1.0 | 275/10 | 1.0 |
| ≤ 10 years | 79/51 | **1.70 1.04 to 2.79**[a] | 79/43 | 1.51 0.9 to 2.55[e] | 79/18 | **1.94 1.01 to 3.73**[g] | 79/12 | **3.43 1.40 to 8.39**[j] | 79/8 | **3.42 1.17 to 9.99**[j] |
| 10+years | 77/54 | **2.12 1.30 to 3.48**[a] | 77/47 | **1.92 1.15 to 3.21**[e] | 77/19 | 1.64 0.88 to 3.09 [g] | 77/11 | **3.03 1.25 to 7.31**[j] | 77/7 | **3.58 1.22 to 10.48**[j] |
| p trend | | < 0.001 | | < 0.001 | | 0.031 | | 0.028 | | 0.023 |
| **Dyes** | | | | | | | | | | |
| Never | 402/179 | 1.0 | 402/157 | - | 402/69 | - | 402/34 | - | 402/22 | - |
| ≤ 10 years | 14/9 | 1.26 0.45 to 3.48[a] | 14/7 | - | 14/3 | - | 14/1 | - | 14/2 | - |
| 10+years | 15/10 | 1.60 0.61 to 4.21[a] | 15/9 | - | 15/3 | - | 15/3 | - | 15/1 | - |
| p trend | 0.208 | | | | | | | | | |
| **Hydrocarbon solvents** | | | | | | | | | | |
| Never | 297/107 | 1.0 | 297/93 | 1.0 | 297/43 | 1.0 | 297/18 | 1.0 | 297/14 | 1.0 |
| ≤ 10 years | 66/43 | 1.65 0.99 to 2.76[a] | 66/38 | 1.57 0.91 to 2.71[e] | 66/16 | 1.87 0.96 to 3.68[g] | 66/10 | **2.84 1.15 to 7.00**[k] | 66/5 | 1.66 0.54 to 5.08 [j] |
| 10+years | 68/49 | **1.91 1.14 to 3.21**[a] | 68/42 | **1.73 1.01 to 2.96**[e] | 68/16 | 1.40 0.73 to 2.70[g] | 68/10 | **2.70 1.12 to 6.48**[k] | 68/6 | 2.47 0.86 to 7.08[j] |
| p trend | | < 0.001 | | < 0.001 | | 0.056 | | 0.053 | | 0.169 |
| **Metals** | | | | | | | | | | |
| Never | 403/179 | 1.0 | 403/158 | - | 403/69 | - | 403/35 | - | 403/21 | 1.0 |
| ≤ 10 years | 13/10 | 1.92 0.65 to 5.72[b] | 13/7 | - | 13/2 | - | 13/2 | - | 13/3 | 3.36 0.77 to 14.71[n] |
| 10+years | 15/9 | 1.22 0.42 to 3.52[b] | 15/8 | - | 15/4 | - | 15/1 | - | 15/1 | 1.47 0.17 to 12.54[n] |
| p trend | 0.176 | | | | | | | | | 0.082 |
| **Pesticides** | | | | | | | | | | |
| Never | 390/156 | 1.0 | 390/137 | 1.0 | 390/59 | 1.0 | 390/28 | 1.0 | 390/19 | 1.0 |
| ≤ 10 years | 20/19 | **2.41 1.13 to 5.13**[a] | 20/16 | 2.15 1.0.96 to 4.84[f] | 20/5 | 1.76 0.60 to 5.14[g] | 20/8 | **4.38 1.61 to 11.93**[j] | 20/3 | **4.04 1.04 to 15.69**[i] |
| 10+years | 10/13 | **2.70 1.01 to 7.21**[a] | 10/10 | 2.38 0.84 to 6.77[f] | 10/5 | 2.82 0.86 to 9.19[g] | 10/2 | 3.05 0.56 to 16.78[j] | 10/3 | **10.65 2.35 to 48.24** [i] |
| p trend | | < **0.001** | | 0.001 | | 0.035 | | < 0.001 | | **0.002** |
| **Sanitizing products** | | | | | | | | | | |
| Never | 242/95 | 1.0 | 242/82 | 1.0 | 242/37 | 1.0 | 242/16 | 1.0 | 242/13 | - |
| ≤ 10 years | 66/40 | 1.15 0.65 to 2.02[c] | 66/34 | 1.17 0.65 to 2.14[f] | 66/15 | 1.73 0.86 to 3.52[i] | 66/9 | 2.00 0.80 to 5.02[l] | 66/6 | - |
| 10+years | 62/49 | 1.41 0.80 to 2.49[c] | 62/43 | 1.47 0.81 to 2.66[f] | 62/17 | 1.76 0.88 to 3.53[i] | 62/11 | 2.21 0.90 to 5.41[l] | 62/6 | - |
| p trend | | 0.001 | | 0.001 | | 0.053 | | 0.052 | | |
| *Biological* | | | | | | | | | | |
| **Biological material** | | | | | | | | | | |
| Never | 403/187 | 1.0 | 403/164 | - | 403/71 | - | 403/37 | - | 403/23 | - |
| ≤ 10 years | 19/7 | 0.35 0.10 to 1.25[b] | 19/6 | - | 19/3 | - | 19/1 | - | 19/1 | - |
| 10+years | 9/4 | 0.78 0.20 to 2.96[b] | 9/3 | - | 9/1 | - | 9/0 | - | 9/1 | - |
| p trend | | 0.656 | | | | | | | | |

*(Continued)*

**Table 3.** (Continued)

| Risk Factors | Total NHL | | B-cell lymphoma | | DLBCL | | Follicular lymphoma | | T-cell lymphoma | |
|---|---|---|---|---|---|---|---|---|---|---|
| | Controls/ Cases | OR adjusted 95% CI | Controls/ Cases | OR adjusted 95% CI | Controls/ Cases | OR adjusted 95% CI | Controls/ Cases | OR adjusted 95% CI | Controls/ Cases | OR adjusted 95% CI |
| **Meat and meat products** | | | | | | | | | | |
| Never | 417/175 | 1.0 | 417/152 | 1.0 | 417/66 | 1.0 | 417/33 | **1.0** | 417/23 | 1.0 |
| ≤ 10 years | 6/11 | **4.16 1.33 to 13.06**[b] | 6/10 | **4.35 1.38 to 13.67**[e] | 6/3 | 3.12 0.69 to 14.19[h] | 6/4 | **13.07 2.79 to 61.12**[j] | 6/1 | 3.64 0.39 to 33.99[j] |
| 10⁺years | 8/12 | **4.19 1.22 to 13.85**[b] | 8/10 | **3.29 1.06 to 10.21**[e] | 8/6 | **4.73 1.37 to 16.73**[h] | 8/1 | 4.27 0.47 to 38.48[j] | 8/1 | 2.92 0.33 to 25.77[j] |
| **p trend** | | **< 0.001** | | **< 0.001** | | **0.001** | | 0.156 | | 0.212 |
| *Physical* | | | | | | | | | | |
| **Industrial radiation** | | | | | | | | | | |
| Never | 411/191 | 1.0 | 411/167 | - | 411/72 | - | 411/37 | - | 411/24 | - |
| ≤ 10 years | 11/4 | 1.04 0.29 to 3.70[d] | 11/4 | - | 11/2 | - | 11/1 | - | 11/0 | - |
| 10⁺years | 9/3 | 0.28 0.05 to 1.52[d] | 9/2 | - | 9/1 | - | 9/0 | - | 9/1 | - |
| **p trend** | | 0.525 | | | | | | | | |
| **Sunlight (occupational)** | | | | | | | | | | |
| Never | 354/136 | 1.0 | 353/116 | 1.0 | 353/51 | 1.0 | 353/24 | 1.0 | 353/20 | - |
| ≤ 10 years | 56/36 | 1.51 0.86 to 2.63[a] | 56/32 | **1.80 1.02 to 3.22**[e] | 56/15 | **2.13 1.02 to 4.47**[h] | 56/9 | **4.14 1.64 to 10.45**[m] | 56/4 | - |
| 10⁺years | 21/26 | 1.91 0.93 to 3.90[a] | 21/25 | **2.47 1.21 to 5.04**[e] | 21/9 | 2.23 0.90 to 5.58[h] | 21/5 | **4.49 1.40 to 14.36**[m] | 21/1 | - |
| **p trend** | | **< 0.001** | | **< 0.001** | | 0.003 | | **0.085** | | |

Total NHL: Total Non-Hodgkin's Lymphoma; DLBCL: Diffuse large B-cell lymphoma; OR: odds ratio; CI: confidence interval.

The data shows missing data.

Any solvents: turpentine, tar, asphalt, benzene, benzine, cleaners or degreaser, synthetic dye hardener, formaldehyde, gasoline, used motor oil, diesel oil, lubricating oil, crude oil, tar, kerosene, reducing agent (not specific), remover (not specs), thinner.

Hydrocarbon solvents: tar, benzene, gasoline, kerosene, thinner (not spec), remover (not spec), thinner.

Metals: chrome, iron, and lead

Pesticides: Pesticides, insecticide, rodenticide, ant killer

[a] OR adjusted for sex, age, place of residence, household income, birthplace, retired, diabetes, cell phone use, wireless phone use, alcohol consumption, HHV8 infection.

[b] OR adjusted for sex, age, place of residence, household income, birthplace, retired, diabetes, cell phone use, wireless phone use, alcohol consumption, HHV8 infection, smoking.

[c] OR adjusted for sex, age, place of residence, household income, birthplace, retired, diabetes, cell phone use, wireless phone use, alcohol consumption.

[d] OR adjusted for sex, age, place of residence, household income, birthplace, retired, diabetes, cell phone use, wireless phone use, HHV8 infection.

[e] OR adjusted for sex, age, place of residence, household income, birthplace, retired, diabetes, cell phone use.

[f] OR adjusted for sex, age, place of residence, household income, birthplace, retired, diabetes, cell phone use, wireless phone use.

[g] OR adjusted for sex, age, place of residence, place of birth.

[h] OR adjusted for sex, age, place of residence, household income, place of birth.

[i] OR adjusted for sex, age, place of birth.

[j] OR adjusted for sex, age, place of residence, diabetes, year of the interview, HHV8 infection.

[k] OR adjusted for sex, age, place of residence, diabetes, year of the interview.

[l] OR adjusted for sex, age, place of residence, year of the interview.

[m] OR adjusted for sex, age, place of residence, diabetes.

[n] OR adjusted for sex, age, retired, cell phone use, alcohol consumption, HHV8 infection.

The categories with models not estimated in the table refer to bivariate analyzes with p>0.25.

Occupations that best reflect exposure to biological materials (nurse, nursing technician)

Occupations that best represent exposure to solar radiation (merchant, domestic worker, electrician, motorist, construction worker, painter, law enforcement officer, security guard, salesman).

—SD = —12.7 years) and controls (mean age: 52.0 years; SD = 12.2). As can be seen, the broad age categorization, the sex, race, and marital status did not affect the comparability of cases and controls. An overview of the demographic characteristics of the NHL case–control are in Table 1.

As can be seen in Table 2, there was no statistical difference in smoking status (p = 0.074) between the two groups, although the smoking burden was higher in the cases than in the controls (> 20 pack years: 26.3% vs 12.3%, p < 0.001). Current alcohol consumption (1 to 5 drinks per week) was lower in the cases than in the controls (26.8% vs 40.0%, p = 0.003). Use of cell phones and wireless phones was more frequent and more prolonged among the controls than the cases (p < 0.001).

Table 3 provides the results from of a multivariate unconditional logistic regression of occupational exposure to chemical, biological, and physical agents as well as NHL development and histological subtypes. The reference group consisted of individuals who were not occupationally exposed to any of these substances/agents.

Exposure to any solvents and pesticides was significantly associated with an increased risk of B- and T-cell lymphoma. We observed a positive dose-response relationship between years of exposure to these substances and increased risk for total NHL (p<0.001).

For occupational exposure to any solvents, hydrocarbons solvents and pesticides, the chance of developing total NHL was 2.12, 1.91 and 2.70 times greater, respectively, in the cases exposed for more than 10 years, compared to the controls (OR = 2.12; 95%CI 1.30–3.48; OR = 1.91; 95%CI 1.14–3.21; OR = 2.70; 95%CI 1.01–7.21) [(see Table 3]).

B-cell lymphoma was significantly associated with occupational exposure to any solvents (> 10 years: OR = 1.92; 95%CI 1.15–3.21) and hydrocarbon solvents (> 10 years: OR = 1.73; 95%CI 1.01–2.96). Risk estimates were stronger for T-cell lymphomas, but wide confidence intervals suggest caution in interpreting them. The chance of developing T-cell lymphomas was 3.58 times greater in the cases exposed for more than 10 years compared to the controls (any solvents > 10 years: OR = 3.58; 95%CI 1.22–10.48).

Occupational exposure to pesticides was significantly associated with an increased risk of for total NHL, follicular lymphoma, and T-cell lymphoma. We observed a dose-response association between years of exposure to these substances and increased risk of total NHL (pesticides>10 years: OR = 2.70; 95%CI 1.01–7.21). Strong associations were found between the development of T-cell lymphoma and exposure to pesticides in the workplace. The chance of developing T-cell lymphomas was 10.65 times greater in the cases exposed to pesticides for more than 10 years, compared to the controls (pesticides>10 years: OR = 10.65; 95%CI 2.35–48.24), The strongest association was observed with risk of T-cell lymphoma, although the small number of subjects suggests caution in the interpretation, due a large confidence interval.

Working with meat and meat products increased 4-fold the risk of total NHL, B-cell lymphoma, and Diffuse large B-cell lymphoma (DLBCL) subtypes (> 10 years: OR = 4.19 95%CI 1.22–13.85; OR = 3.29 95%CI 1.06–10.21; OR = 4.73 95%CI 1.37–16.73, respectively). The small number of subjects also suggests caution in the interpretation.

We observed that the chance of developing B-cell lymphoma was 2.47 times higher in the cases exposed to sunlight for more than 10 years, compared to the controls (OR = 2.47 95%CI 1.21–5.04). Overall, the chance that those occupationally exposed for at least 10 years to sunlight had NHL was more than double compared to the controls. Follicular lymphoma was significantly associated with occupational exposure to sunlight (> 10 years: OR = 4.49 95%CI 1.40–14.36).

No statistically significant associations were found between some exposures at work (dyes, metals, sanitizing products, biological material, and industrial radiation) and NHL.

## Discussion

Most of the studied NHL cases were diagnosed as B-cell lymphoma, with a higher incidence of diffuse large B-cell lymphoma (DLBCL) (82; 38.1%) and follicular lymphoma (40; 18.6%). This corresponds to the incidences found in the literature for the different histological subtypes of NHL, which are the high-grade and aggressive subtypes [1, 19].

In this study, we found an association between NHL and the following occupational exposures: any solvents, hydrocarbons solvents, pesticides, meat and meat products and sunlight, when compared to those not occupationally exposed to these factors. These associations were maintained even after adjusting for variables that could confound (described at the bottom of Table 3).

A significant upward trend with years of exposure was detected for any solvents and hydrocarbon solvents (NHL (any subtype), B-cell lymphoma, and T-cell lymphoma), pesticides (NHL (any subtype), and T-cell lymphoma), meat and meat products (NHL (any subtype) and DLBCL), and sunlight (B-cell lymphoma ). No associations were found between NHL and exposure to dyes , metals, sanitizing products, biological material, and industrial radiation (of any sub-types).

Among the categories of solvents presented in the occupational exposure questionnaire questions, benzene was one of the chemical agents most reported by study participants. In Brazil, according to Corrêa et al (2016) [20], 7.376,761 (8.5%) workers belong to occupational groups potentially exposed to benzene, while 770,212 workers are considered definitively exposed to benzene. Occupational exposure to benzene and other aromatic hydrocarbons and the possibility of developing NHL have been investigated in other works. An Italian study by Miligi et al. [21], which evaluated 1,428 cases of NHL and 1,530 controls in eight areas of Italy, found a positive association for exposure to benzene (OR: 1.60; 95%CI 1.0–2.4) and aromatic hydrocarbons (OR: 1.20; 95%CI 0.9–1.7). A meta-analysis found in 22 studies of benzene exposure a 22% increased risk of NHL (OR: 1.22; 95%CI 1.02–1.47). When the estimate was calculated excluding the based studies only in the self reported occupational history, with 6 studies remaining, the risk was approximately 2 times higher for NHL (OR: 2.12, 95%CI 1.11–4.02). [22]. Other aromatic hydrocarbon solvents (styrene, BTX, toluene, xylene, and others) have been associated with NHL [23–25]. It is worth mentioning that workers are rarely exposed to a single agent, since workplaces containing benzene, also expose them increase exposure to a complex mixture of other aromatic hydrocarbons, such as toluene, xylene, and ethylbenzene; as well as other hydrocarbons derived from petroleum during the complex processes arising resulting from petroleum refining, the production of coke, oils and naphtha [26].

The wide presence of heavy metals in occupational, industrial, and agricultural environments leads to increased occupational exposure and concern about their health effects [27]. In the present study, no association was found between metal exposure and overall NHL or its subtypes, although a significant odds ratio was observed for overall NHL (OR = 1.92 CI: 0.65–5.72 for those exposed < = 10 years) and for follicular lymphoma (OR = 1.22 CI: 0.42–3.52 for those exposed > = 10 years), albeit without statistical significance. The same was observed for follicular lymphoma (OR = 3.36 CI: 0.77–14.71 for those exposed < = 10 years and OR = 1.47 CI: 0.17–12.54 for those exposed > = 10 years), also without statistical significance. This result is consistent with a study conducted in Australia that similarly assessed the risk of developing NHL in individuals with occupational exposure to metal dust, which did not show a positive association [15], but Kelly et al [28] conducted a study in Italy and Sweden that observed a positive association in women between heavy metal levels in the blood and the risk of developing NHL. Data in the literature are still inconsistent and scarce; nevertheless, the results of this

study found a positive association between exposure to metals and the development of certain types of NHL. Subsequent studies are needed to clarify this association.

In the present study, an association was found between exposure to pesticides and the development of NHL. Case-control, cohort, and meta-analysis studies have explored the association between exposure to active ingredients of specific pesticides and development of the NHL development [29]. The IARC classified some pesticides as exhibiting carcinogenic potential after observing a positive association between pesticides exposure and NHL development. DDT, diazinon, glyphosate, and malathion have been classified as probably carcinogenic to humans (group 2A) [30]. Furthermore, 2,4-D, chlordane, heptachlor, hexachlorobenzene, lindane, mirex, and pentachlorophenol were classified as possibly carcinogens to humans (group 2B) [29]. Glyphosate and 2,4-D are among the most used pesticides in Brazilian crops. Leon et al. [31] found an association between NHL and occupational exposure to pesticides in cohorts from France (Agriculture and Cancer–AGRICAN); Norway (Cancer in the Norwegian Agricultural Population–CNAP); and the United States of America (Agriculture Health Study–AHS). For most of the chemical groups in the pesticides evaluated, no associations were observed in 33 active ingredients. However, increased total NHL risk was observed in farmers who used terbufos; CCL/SLL among those who used deltamethrin; and DLBCL among those who used glyphosate. It should be clarified that terbufos is an extremely toxic organophosphate insecticide; deltramethrin is an insecticide of the pyrethroid group, and glyphosate is an herbicide. All three are widely used in Brazil, including in unauthorized cultures crops.

The results of this study also showed the association of exposure to meat and meat products with an increased chance of developing NHL. These results corroborate the results those found in the case-control study conducted in eight Canadian provinces by Fritschi et al. [32], who which reported an increased risk a link between exposure to beef cattle and NHL; as well as by they also corroborate the results in Tranah et al [33], who observed an increased risk of NHL in individuals who lived or worked with cattle (1.6, CI = 1.0–2.5) and pigs (OR = 1.5; CI = 1.1–2.1). With exposure to pigs for $\geq$ 5 years, the increased the risk also occurred for NHL (OR = 1.8; CI = 1.2–2.6) and for diffuse large-cell and immunoblastic large-cell lymphoma (DLCL) (OR = 2.0, CI = 1.2–3.4). According to the authors, contact with infectious agents as a result of work with livestock or contact and handling of pork and beef increases the risk of NHL, as they can carry lymphomagenic infectious agents. Freeman et al. [34] analyzing data from the Agricultural Health Study, observed increased risk of NHL (RR = 1.6, 95% CI 1.0–2.4) among those who raised poultry and those who were veterinarians on poultry farms (RR = 12.2; 95% CI:1.6–96.3) [33].

In New Zealand, Mannetje et al. [8] observed a 2-fold higher risk of NHL that is two times larger in among workers who handled meat, especially those who worked at meat processing plants, when compared to controls. Although the studies described found some association between exposure and NHL, this association was not for all investigated animals or for all NHL subtypes. Freeman et al. [34] for example found an association for between raising poultry and increased risk for NHL, but not for sheep; Fritschi et al. [32] found an association between NHL and DLCL for occupational exposure to beef cattle, but not to any other animal.

Our study indicated that occupational exposure to sunlight for at least 10 years doubled the chance of B-cell lymphomas, DLBCL, and Folicular Lymphoma compared to the controls. Lu et al. [14] in a meta-analysis with 10 case-control studies and one cohort, in the period between from 1997 and to 2012, observed a positive association between occupational sunlight exposure and NHL (OR = 1.15; 95%CI 1.05–1.23), but not for the common subtypes of non-Hodgkin's lymphoma (T-cell and B-cell lymphoma, and chronic lymphocytic leukemia). Similarly, Karipidis et al. [35] observed that the group with cumulative occupational exposure and a minimum exposure time of 5 years before diagnosis had been a 1.46 times-fold higher more likely

chance of to develop having NHL (95%CI 1.06–2.02) than the unexposed. Although the association between occupational sunlight exposure and NHL was mentioned in the studies described here, other studies have not found this the same association [36, 37]; therefore, more research is needed.

The authors considered it important to evaluate the association between subtypes of non-Hodgkin lymphomas and occupational exposures, since few articles carried out this investigation. However, for some types of occupational exposures, such as biological material, dyes, metals, biological material, and industrial radiation, as the number of observations was very small, it was not possible to perform statistical analysis by subtypes of non-Hodgkin's lymphoma. Therefore, only the results of the main outcome, total NHL, were evaluated. Furthermore, no signs of effect modification were observed when measures of association were compared for the strata of each variable for total NHL and for the different subtypes of non-Hodgkin's lymphoma.

Regarding sociodemographic characteristics of the participants, men exhibited a higher frequency of NHL cases. This demographic profile agrees with is in line with estimates of the Brazilian National Cancer Institute [7] and with the profile of cases described in the literature. The mean age of cases was 56 years old (SD = 12.7 years), with the largest portion of individuals over 45, in accordance with according to the InterLymph consortium [37–40]. It is noteworthy that the broad age categorization did not affect the comparability of cases and controls by gender and case-control status.

Both cases and controls reside are in the metropolitan mesoregion of the Rio de Janeiro state. This data has also a limitation. INCA is a reference hospital in cancer and one of the only ones to offer comprehensive and free treatment for cancer (public health), patients from all over the state of Rio de Janeiro want to be treated at INCA. Therefore, some individuals, from other regions, outside the metropolitan region, came to INCA. In addition, it is important to emphasize that most individuals stop working, retire, or take a leave of absence after discovering the disease, which lowers the family income. This could be one of the explanations for the lower income of the cases compared to the controls [41].

For follicular and marginal zone lymphoma, smoking is a risk factor [42, 43]. The immunological effects observed in smokers, such as immunosuppression and increased leukocyte production, may be associated with the development of lymphomas [44]. In addition, cigarette smoke contains substances known to be carcinogens, including benzene and formaldehyde [30]. In a cohort study conducted in the United States with 1,926 cases of NHL, smoking (including former smokers) was associated with the occurrence of NHL (histological type of T-cells) [45]. The present study found no statistical differences in smoking between the two groups, although the smoke load (pack years) was higher among the cases than the controls. As It is noteworthy that many patients stop smoking after being diagnosed with cancer. Therefore, their answers given by patients when asked about regarding being current smokers may reflect the condition of no longer smoking them having stopped soon after the diagnosis of NHL.

No increase in alcoholic beverages consumption of alcoholic beverages was observed among the cases after analyzing the information on the number of doses of alcoholic beverages consumed per week; therefore, the frequency found in the present study (61%) was lower than in other studies [46].

## Limitations of the study

The authors consider the presented results to be of utmost relevance considering the history of intensive use of carcinogenic agents classified in Group 1 by IARC in the country Brazil, some of which have already been banned in other countries. It should also be noted that there is a scarcity of analytical case-control studies in Brazil that include occupational risk factors for

cancer. Another important issue is the possibility of these results providing tools for occupational health surveillance in states and municipalities regarding workers' exposure to the agents studied here. However, we acknowledge that these results should be viewed with caution, considering some problems inherent to the study design itself.

The authors attempted to minimize the potential selection bias of controls by recruiting companions of hospitalized patients diagnosed with solid cancers. As this group was predominantly composed of women, another recruitment strategy was used. During the visits of relatives, friends, and neighbors of hospitalized patients, they were routinely invited to become blood donors. Upon acceptance, they were invited to participate in the research. Only blood donors who went to visit hospitalized patients were recruited. Thus, the blood donors did not differ significantly from the controls recruited in the hospital wards. It is evident that those who agreed to donate blood may differ slightly from those who did not donate, in terms of their health status, considering that the donors are likely to be healthier individuals who are still relatives, neighbors, and/or friends. It should also be noted that INCA is not a conventional blood collection institution like such as a regular blood center.

Another issue that should be addressed as a potential limitation for the findings is the multiple comparisons involved in this study. This is an exploratory study that sought to evaluate various exposures in a single study. For this reason, important details such as occupation, economic activities, and exposures could not be assessed in a more disaggregated manner, limiting the observation of effects in a more specific way. It was also not possible to investigate multiple simultaneous exposures.

It should also be considered as a limitation is the inherent memory bias in the study design. In this case, it was further accentuated exacerbated by the administration application of an occupational history questionnaire to capture the recall of occupational history, which was necessary for analyzing the association between work and NHL in this study. Each patient was asked to report all occupations jobs held in over the last past 20 years. Cases may have made more effort to remember possible exposures compared to controls.

It is also noteworthy that, the study did not include any environmental assessment or clinical evaluation of cases and controls. The only tool used was a data collection instrument, and the information was obtained through interviews. In other words, we only have self-reported exposures mentioned by the interviewees. In an attempt to observe cumulative exposure, only years of exposure to certain carcinogenic agents were considered. There was no way to measure the intensity of each exposure.

It is important to emphasize that the study was carried out in a National Cancer Institute of the Brazilian Ministry of Health of Brazil, which is a reference center for cancer, and is located in the state of Rio de Janeiro, which is a very populous state in Brazil, it is assumed that the sample included in the study is representative of the general population and its results can be extrapolated to the national level and even to other countries with the same socio-demographic and occupational characteristics as Brazil.

## Conclusions

The data from the present study paper support the hypothesis that occupational exposure to any solvents, hydrocarbons solvents, pesticides, meat and meat products, and sunlight is related to an increased chance of developing non-Hodgkin's Lymphoma in adults. However, occupational exposure to dyes, sanitizing products, biological materials, metals, and physical industrial radiation was not statistically associated with this type of neoplasm. The results of this study corroborate other international studies that evaluated and found an association between some occupational exposures and NHL.

However, these results must be viewed with caution, since the results were presented by the total number of cases and by subtypes of lymphomas, reducing the number of observations in some cells during the crossing with exposure groups when cross-examined against exposure groups. This form of analysis generated yielded measurements of great magnitude for some subtypes, but with wide confidence intervals. Therefore, some without measurements lack statistical significance, requiring is careful inspiring care in the evaluations and conclusions. It should also be noted that the study was an exploratory study, which and that it allowed assessing which occupational risks were associated with NHL in Brazil. From this preliminary study we will be able to advance make some progress in specific exposures. In several developed countries or in countries with the same socio-demographic and occupational characteristics as Brazil, some agents' substances evaluated here are used intensely, requiring constant monitoring and surveillance of these regarding exposures, exposed workers, and diseases, including cancer, such as NHL. Therefore, the results of this present study can contribute to and support strategies in for public health and surveillance programs for workers. This paper will also help the various institutions in the world involved in public health actions worldwide to design strategies for the prevention and control of work-related cancer.

## Acknowledgments

Adriana Scheliga from the Oncology Service of the Brazilian National Cancer Institute (INCA); Employees of Hematology Service and National Tumor Bank (BNT) of the Brazilian National Cancer Institute (INCA).

## Author Contributions

**Conceptualization:** Marcia Sarpa, Vanessa Índio do Brasil da Costa, Claudio Gustavo Stefanoff, Rocio Hassan, Ubirani Barros Otero.

**Data curation:** Marcia Sarpa, Vanessa Índio do Brasil da Costa.

**Formal analysis:** Vanessa Índio do Brasil da Costa, Arthur O. C. Schilithz.

**Funding acquisition:** Marcia Sarpa, Ubirani Barros Otero.

**Investigation:** Marcia Sarpa, Vanessa Índio do Brasil da Costa, Sâmila Natiane Ferreira, Carolina Ávila de Almeida, Paula Gabriela Sousa de Oliveira, Letícia Vargas de Mesquita, Rocio Hassan, Ubirani Barros Otero.

**Methodology:** Marcia Sarpa, Vanessa Índio do Brasil da Costa, Sâmila Natiane Ferreira, Carolina Ávila de Almeida, Paula Gabriela Sousa de Oliveira, Letícia Vargas de Mesquita, Arthur O. C. Schilithz, Claudio Gustavo Stefanoff, Rocio Hassan, Ubirani Barros Otero.

**Project administration:** Marcia Sarpa, Ubirani Barros Otero.

**Resources:** Marcia Sarpa.

**Supervision:** Marcia Sarpa, Vanessa Índio do Brasil da Costa, Ubirani Barros Otero.

**Validation:** Marcia Sarpa.

**Visualization:** Marcia Sarpa.

**Writing – original draft:** Marcia Sarpa, Vanessa Índio do Brasil da Costa, Sâmila Natiane Ferreira, Carolina Ávila de Almeida, Paula Gabriela Sousa de Oliveira, Letícia Vargas de Mesquita, Arthur O. C. Schilithz, Rocio Hassan, Ubirani Barros Otero.

**Writing – review & editing:** Marcia Sarpa, Vanessa Índio do Brasil da Costa.

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
