## [Decision Letter · Decision Letter 0]

1 Dec 2022

PONE-D-22-29162INVESTIGATION OF OCCUPATIONAL RISK FACTORS FOR THE DEVELOPMENT OF NON-HODGKIN LYMPHOMA IN ADULTS: A HOSPITAL-BASED CASE-CONTROL STUDYPLOS ONE

Dear Dr. Marcia Sarpa,

Thank you for submitting your manuscript to PLOS ONE. After careful consideration, we feel that it has merit but does not fully meet PLOS ONE’s publication criteria as it currently stands. Therefore, we invite you to submit a revised version of the manuscript that addresses the points raised during the review process.

We look forward to receiving your revised manuscript.

Kind regards,

Elingarami Sauli, PhD

Academic Editor

PLOS ONE

Reviewers' comments:

Reviewer's Responses to Questions

**Comments to the Author**

1. Is the manuscript technically sound, and do the data support the conclusions?

Reviewer #1: No

Reviewer #2: No

2. Has the statistical analysis been performed appropriately and rigorously? 

Reviewer #1: No

Reviewer #2: No

3. Have the authors made all data underlying the findings in their manuscript fully available?

Reviewer #1: Yes

Reviewer #2: No

4. Is the manuscript presented in an intelligible fashion and written in standard English?

Reviewer #1: Yes

Reviewer #2: No

5. Review Comments to the Author

Reviewer #1: The paper is well written. However, unfortunately, there are major biaises in the inclusion of cases and controls. Since controls differ largely from cases, it is very hard to draw conclusions from this study.

More cases reside in other regions than controls (20% vs. 5.7%, p < 0.001). The cases had a low

socioeconomic status with a lower educational level (47.2% vs 27.3%, p < 0.001) and family

income (mean minimum wage: 2,763.5 BRL vs 4,303.6 BRL, p > 0.001) than the controls.

The cases included more retirees (28.4% vs 21.4, p = 0.046), pensioners (6.5% vs 2.9%, p =

0.025), and those who received financial assistance from the government (27.4% vs 1.5%, p

222 < 0.001) than in the controls.

Reviewer #2: In the last 45 years a number of case-control studies have investigated in detail the risk factors of non-Hodgkin Lymphoma (NHL). In 2001, the U.S. National Cancer Institute promoted the InterLymph Consortium between the research groups conducting studies worldwide. Great achievements have followed in terms of the identification of environmental, behavioural, clinical, and genetic conditions predisposing to the appearance of NHL. The interdisciplinary collaboration within InterLymph has allowed to define specific risk patterns for individual Lymphoma subtypes, although some confusion still persists because of the difficulty in matching the progress in the classification, diagnosis, and treatment of these diseases with the standardization and statistical power requirements of Epidemiology. Unfortunately, studies from various parts of the world, such as Asia, Africa, Middle East, and South America, are not part of InterLymph, because of not matching the requirements for participation or simply because there are no studies out there. Therefore, this reviewer welcomes this Brazilian case-control study and hopes the authors will continue their efforts and do their best to join the InterLymph Consortium.

This is a relatively small size study, which would benefit of a more detailed description of the methods and the classification scheme adopted to define the disease entity. There is discussion of the limitations, and the conclusion overinflates the positive findings.

On page 6, row 137, after the sentence ending with “...the diagnosis of NHL.” Please explain whether the diagnosis was based on the WHO classification, and, if so, which revision, or was it the International Classification of Diseases -10th revision. The C82-85 codes reported on page 6, row 135, should indicate that they refer to the ICD-10.

The matching criteria should also described. On page 6, row 137, the authors wrote “The studied cases were paired by sex and age group with 455 healthy individuals (controls)”. First, the correct term would be “approximately 2:1 matched”. Second, within what age range was matching acceptable? Were they 5-year or 10-year age-groups? From Table 1, seems that it was 20–year, which would be quite large. However, based on the small difference in the mean age between cases and controls reported on page 9, rows 212-214, the matching was effective in making cases and controls comparable by age. So, please, explain in the discussion that the broad age categorization did not affect the comparability of cases and controls by gender and case-control status. Third, on rows # 141-142, the authors wrote that “...for each new case diagnosed and included in the study, a new control was chosen...”. Well, the controls were twice as the cases, so at least two controls has to be matched to each case. Fourth, the 455 controls were presumably “healthy” not “all healthy”, as only those with a history of cancer were excluded.

The source of the data is unclear. The rows 167-181 on page 7 describe a list of data that were “collected. A questionnaire and interviews are mentioned but was it a structured interview with a standardized questionnaire? Was the questionnaire administered in person by trained nurses/interviewers or self-administered? Was it a consent form signed before the interview?

Please note that, on page 6 row 150, the number of participating controls is 577, while it is 557 in the flow diagram in figure 1. In the diagram and in the text the refusals to participation are not indicated. While aware of the higher participation rate of hospital controls vs population controls, this reviewer does not believe that every one of the eligible controls accepted. The 20 missing in the count (3.5%) are proportionally identical to the refusals among the cases (which, by the way, is 3.6 not 3.5%).

The occupational information was limited to the main occupation in the last 20 years lasting one year or more. As this article mainly focuses on occupational risk factors, this reviewer would have expected a more detailed work history. Those unaware of the Brazilian labour market situation, as this reviewer is, cannot judge whether a substantial part of short-term, low-paying and possibly highly exposed jobs were excluded from study. Also, who did assess the exposures? Where these self-reported by the study participants? Was an expert occupational physician/industrial hygienist involved in the exposure assessment? Were jobs and industry coded in a standard format? Was a job-exposure matrix applied to the occupation and industry codes of study subjects? Apparently not, as the generic definitions of solvents, paints, metals, pesticides, cleaning products, biological material, animal and viscera material, industrial radiation and solar radiation seem to suggest. The current standards of occupational epidemiology research are far ahead in respect to this stage. Please, describe this as a limitation in interpreting the findings from this study. Besides, a little background of what occupations contributed to the exposure to solvents and metals might help the interpretation. Likewise, the prevalent type of crops and some information on the best selling pesticides (insecticides, herbicides and fungicides mainly) in the study area, and not just overall Brazil, might also be helpful.

On page 7, rows 198-199, the authors wrote: “By unconditional logistic regression, multivariate models were estimated to analyze exposure...” This is an awkward sentence. The authors refer to the use of unconditional logistic regression to estimate the Odds Ratio for specific disease entities associated with the exposures under scrutiny. However, the miracle of multivariate analysis is to adjust the risk estimate of the covariate of interest by the concurrent effect of other covariates. Usually age, gender, and education are included by default. Other covariates might be relevant, such as income, marital status, and residence whether in metropolitan Rio de Janeiro or elsewhere. So, please, list what covariates were included in the logistic model and whether a forward or backwards stepwise procedure was adopted to select them. It is the personal opinion of this reviewer that, in the omics era, race categories based on the colour of the skin do not make any sense and are not surrogate of anything. Therefore, he would suggest to omit it from Table 1 , from page 9, rows 214-215, and from page 8 rows 304-311, or to justify the use of this otherwise useless category.

Table 2 indicates that three controls and one case had a diagnosis of myelodysplasia. Myelodysplastic syndromes were so defined for the first time by the French-American-British Working Group in 1976, and considered as a precursor stage for acute myeloid leukaemia. This seems odd, as the authors declared to have excluded subjects with any form of cancer or visiting parents affected by haematological diseases from the control group. Please, justify the inclusion of the myelodysplasia cases or rerun the analysis after excluding them.

Cases of follicular lymphoma were 40 and those of T-cell lymphoma were 26. Why Table 3 shows results for the second and DLBCL, but not for follicular lymphoma? Because of the recent observation of the elevated risk of follicular associated with use of glyphosate, and the note in the discussion about glyphosate being one of the most used pesticides in Brazil, perhaps and analysis of the 40 cases of follicular lymphoma in relation to pesticide use might be interesting.

When presenting the results, the authors gave too much emphasis to the positive associations with risk of T cell lymphoma, which were generated by small numbers. An introductory sentence such as “Risk estimates were stronger for T-cell lymphomas, but the wide confidence intervals suggest caution in interpreting them.” In particular, the 5-fold excess risk of T-cell lymphomas associated with exposure to metals suffers from three drawbacks: first the generic definition of the risk factors, second the extremely wide confidence interval, and third the inverse dose-response trend with years of exposure. Likewise, the almost 17-fold excess risk of T-cell lymphoma associated with generic exposure to pesticides seems to be mainly due to the wild ups and downs in the risk estimates generated by the small number of the exposed for 10 years or more. As it concerns the sanitizing products, this reviewer is puzzled by not finding a T-cell lymphoma risk estimate in Table 3, associated with the and 7 cases exposed for less or more than 10 years.

This reviewer also suspects that the association with exposure to solar radiation was mostly, if not entirely, driven by farmers. Were there any non-farmers exposed to solar radiation (fishermen for instance or any other open-air occupations)?

The first part of the discussion illustrates the results of lifestyle (smoking and alcohol), and clinical variables (diabetes, viral infections) that were generated by univariate n-by-two tables and not by the multivariate analysis. There are also several drawbacks in the way these were classified; for instance the cut-off for being classified as an alcohol drinker was ridiculous (one drink per week). Considering the U-shaped curve of its relationship of alcohol-related cancer and other diseases, perhaps a categorization of drinkers as occasional and regular would have been more meaningful. Besides, there is no distinction between type 1 and Type 2 diabetes, which is mostly a feature of the metabolic syndrome. Viral diseases, including HTLV infection, were apparently self reported (there is no mention of serological testing). However, while there is no discussion about the more numerous self-reported cases of HCV, a disproportionately long paragraph is dedicated to discuss the HTLV finding. Both would be worth a short mention with proper caution because of the small study size, the small number of positive answers, and the crude testing of the difference. A long discussion on benzene follows without explaining whether benzene and the other aromatic hydrocarbons were part of the category of hydrocarbon solvents. In fact, such a definition would include also the widely used aliphatic hydrocarbons (such as n-hexane) and the less refined petroleum-derived solvents, such as mineral spirit, gasoline, and naphta. Not to mention the weak finding on metals and T-cell lymphoma or that on solar radiation. The whole discussion needs to be reshaped and shortened. The limitations (small study size, occupational history limited to the longest most recent job, reliance on self reported exposures, generic definition of the relevant occupational risk factor, multiple comparisons) need to be properly highlighted and discussed.

The conclusion, instead, overemphasize the observed positive associations.

Minor points

Introduction, page 3, rows 81-83: “NHL is among the top ten most common types of cancer in Brazil, each year, for the triennium 2020-2022. An estimated 6,580 and 5,450 new cases occur annually in Brazilian men and women”.

The triennium 2020-2022 is not over yet. So, the authors might rephrase this sentence. This reviewer suggests something like “...common types of cancer in Brazil, each year. For the triennium 2020-2022, an estimated 6,580 and 5,450 new cases will occur”.

Introduction, page 4, row 84: this is a “rate” not a “risk”. Please, use the correct term.

Introduction, page 4, rows 97-100: “Other occupational factors related to NHL include solar radiation (12); ionizing radiation (11); organic dust (grain, fabric, and wood dusts); and mineral and metal dust; however, evidence of the association of these factors with NHL is not yet consistent (7).”

Well, these factors have been investigated, but the evidence for their association with an increase in risk is not simply “not consistent”: it does not exist. For instance, the reference # 12 found an inverse association between exposure to solar radiation and NHL risk. Note that reference # 11 refers to agricultural work not ionizing radiation; a range of suitable citations or a review (for instance Schmitz-Feuerhake I et al. Ann Hematol. 2022;101(2):243-250) might better serve the purpose. Also, a citation might accompany the mention of organic dust (Cocco P et al. Scand J Work Environ Health 2021;47(1):42-51) and mineral and metal dust (Fritschi L et al. Cancer Causes Control. 2005;16(5):599-607).

Methods, page 5, rows 118-119. The comparison with Portugal is unnecessary. Please, delete this sentence.

Methods, page 5, row 129. This reviewer is unaware of the need of establishing the sampling error in calculating the statistical power of a case control study. Please make a reference or omit mentioning the sampling error.

Methods, page 5, row 132. “OR = 2,5”. Please, use the period to separate the decimal.

Page 6, methods, rows 144-154. Please move these two paragraphs immediately after the statistical power paragraph and before indicating the final number of cases included in this study. Please, replace the terms “That way...” with “Therefore...”.

Page 6, row 144: the verbal form “were” should be included after “...for [the] cases...”

Page 7, row 174. What is the point of mentioning the minimum salaries? Wouldn’t the family income be enough? How was the amount of 5BRL selected for the cut-off? Does it make any sense showing also the mean income in Table 1?

Page 7, row 176. “cellular use” perhaps refers to cellular phone. Please, rephrase.

Page 8, row 195. “Missing values were excluded from the analysis”. Table 1 should indicate how many missing values were there for each variable of interest.

Page 8, Results, row 209. “...and T-cell lymphoma (30; 14.0%)”. As it is written seems the authors are saying that T-cell lymphomas were also prevalent. The sentence should be written as it follows “...and T-cell lymphomas were 30 (14.0%).”

Page 11, rows 234-235: “We also found a difference in the use of cell and wireless phones (p > 0.001)” and Table 2. The sentence is formulated ambiguously. In fact, the point is that controls used cell phones and wireless phones more than the cases, which might be a consequence of the lower income and lower education of the cases in respect to the controls. Therefore, the phrasing should be explicit about the fact that “use of cell phones and wireless phones was more frequent and more prolonged among the controls than the cases”.

Page 13, row 254, same page, row 272 : use the terms “years of exposure” instead of “exposure time”.

Page 18, row 303: delete the phrase “by histological subtype, and these data are”

Page 18, rows 304-311. Delete the whole sentences.

Page 18, rows 314-315: “Among the NHL cases, some patients resided in other regions of Rio de Janeiro.” Unclear statement. Does it mean that a proportion of the cases larger than controls resided in regions other than metropolitan RJ?

Page 19, row 319. Delete the colloquial term “your” and replace it with the article “the”.

Page 19, rows 321-327. Smoking is not a strong risk factor for lymphoma, apart from a weak association with risk of follicular lymphoma. Besides, the case-control difference in the proportion of smokers was not the result of the multivariate analysis but of a simple 2x2 table, and current smokers prevailed among the controls. This reviewer suggest the authors to stick to their findings and avoid discussing meaningless findings.

Please, format the list of references in a consistent way following the journal guidelines.

6. PLOS authors have the option to publish the peer review history of their article (what does this mean?). If published, this will include your full peer review and any attached files.

Reviewer #1: **Yes: **Caroline Besson

Reviewer #2: No

---

## [Author Response · Author response to Decision Letter 0]

25 Mar 2023

PONE-D-22-29162

INVESTIGATION OF OCCUPATIONAL RISK FACTORS FOR THE DEVELOPMENT OF NON-HODGKIN LYMPHOMA IN ADULTS: A HOSPITAL-BASED CASE-CONTROL STUDY

PLOS ONE

Answer: All participants/volunteers in the case-control study signed a free and informed consent form, agreeing to participate in the research before answering the questionnaires.

The study did not include minors.

Answer: We can provide all data and study information requested by PlosOne. We have the applied questionnaires, the research ethics committee approval term, the informed consent form and all the analyzes carried out.

Answer: We made a mistake when writing “data not shown”, because in reality this data was presented in the manuscript. We can send you an excel spreadsheet with all the information on the histological types of lymphomas included in the study if you wish.

 

RESPONSE TO REVIEWERS

5. Review Comments to the Author

Reviewer #1: The paper is well written. However, unfortunately, there are major biases in the inclusion of cases and controls. Since controls differ largely from cases, it is very hard to draw conclusions from this study.

More cases reside in other regions than controls (20% vs. 5.7%, p < 0.001). The cases had a low

socioeconomic status with a lower educational level (47.2% vs 27.3%, p < 0.001) and family

income (mean minimum wage: 2,763.5 BRL vs 4,303.6 BRL, p > 0.001) than the controls.

The cases included more retirees (28.4% vs 21.4, p = 0.046), pensioners (6.5% vs 2.9%, p =

0.025), and those who received financial assistance from the government (27.4% vs 1.5%, p

222 < 0.001) than in the controls.

Answer: We appreciate the review and suggestions given by this reviewer, which were extremely important to improve the article.

The study carried out was a hospital-based case-control and the basic principle of selection of controls in this type of study is that they need to represent the base population from which the cases came. (Rothman K.J et al. Modern Epidemiology, page 137, paragraph 4). Although cases and controls in this study are different in some social and economic aspects, visitors and/or companions of patients generally come from the same population of origin of the cases and, therefore, have the same chance of being included among the cases and seeking care at the selected hospital if they were to develop cancer. This is justifiable because INCA is a reference institution for cancer treatment in the state of Rio de Janeiro and in Brazil, since most municipalities in the state of Rio de Janeiro do not have hospitals for cancer treatment. In this way, cancer patients within the public health system of Rio de Janeiro, Brazil, are referred to the National Cancer Institute.

Furthermore, the data in Table 1 were reviewed and there are no statistically significant differences for the main variables between the control and case groups. Place of residence and family income were considered for adjustment in the multivariate models with the aim of adjusting the model by controlling the effect of these variables individually.

The limitation regarding these two variables in the discussion was also described. In both groups, individuals reside in the metropolitan mesoregion of the state of Rio de Janeiro. This data has a limitation, because many individuals can omit the correct place of residence.

Regarding the employment status variables (retired, pensioners and those who received financial aid from the government), it is known that many patients diagnosed with cancer and undergoing cancer treatment in Brazil stop working, retire, or become pensioners and receive financial aid from the government.

It is noteworthy that the use of multivariate analysis models proposes to control and weigh all these differences, whether they are collected through medical records or through direct questions asked using forms. 

Reviewer #2: 

1) In the last 45 years a number of case-control studies have investigated in detail the risk factors of non-Hodgkin Lymphoma (NHL). In 2001, the U.S. National Cancer Institute promoted the InterLymph Consortium between the research groups conducting studies worldwide. Great achievements have followed in terms of the identification of environmental, behavioural, clinical, and genetic conditions predisposing to the appearance of NHL. The interdisciplinary collaboration within InterLymph has allowed to define specific risk patterns for individual Lymphoma subtypes, although some confusion still persists because of the difficulty in matching the progress in the classification, diagnosis, and treatment of these diseases with the standardization and statistical power requirements of Epidemiology. Unfortunately, studies from various parts of the world, such as Asia, Africa, Middle East, and South America, are not part of InterLymph, because of not matching the requirements for participation or simply because there are no studies out there. Therefore, this reviewer welcomes this Brazilian case-control study and hopes the authors will continue their efforts and do their best to join the InterLymph Consortium. This is a relatively small size study, which would benefit of a more detailed description of the methods and the classification scheme adopted to define the disease entity. There is discussion of the limitations, and the conclusion overinflates the positive findings.

Answer: Thanks for the review and suggestions given by this reviewer. We believe that they were fundamental for the clarification of the work and for the improvement of the article.

As requested, we inserted a paragraph in the conclusion, with the aim of reducing the weight of the positive findings.

It is also important to emphasize that the study was exploratory in nature and that it is one of the first studies in Brazil with this objective. The study made it possible to assess which occupational risks were associated with NHL in Brazil. From this preliminary study we will be able to advance and evaluate specific/detailed expositions. It should also be noted that this article presents the first results of this study, in a more global way, and that we intend to better address occupations, economic activities and the development of non-Hodgkin's lymphoma in another article.

We put in the conclusions that “the results were presented by the total number of cases, but also by subtypes of lymphomas, reducing the number of observations in some cells during the crossing with exposure groups, which generated measurements of great magnitude for some subtypes, but with wide confidence intervals, and some without statistical significance, requiring parsimony in evaluations and conclusions”.

We also include a more detailed description of the methods and classification scheme adopted.

2) On page 6, row 137, after the sentence ending with “...the diagnosis of NHL.” Please explain whether the diagnosis was based on the WHO classification, and, if so, which revision, or was it the International Classification of Diseases -10th revision. The C82-85 codes reported on page 6, row 135, should indicate that they refer to the ICD-10.

Answer: The patients (cases) eligible for the present study were: patients diagnosed with a primary tumor of non-Hodgkin's Lymphoma (NHL), with histological confirmation by immunophenotyping, according to the anatomical classifications of the international code of diseases (ICD-10), as described below: Follicular non-Hodgkin's lymphoma (C82); Diffuse non-Hodgkin's lymphoma (C83); Cutaneous and peripheral T-cell lymphomas (C84); Non-Hodgkin's lymphoma of other and unspecified types (C85) (WHO, 2008). Patients diagnosed with HIV were not eligible for the study, as this syndrome is associated with the development of histological subtypes of NHL (CHIU et al., 2003). The diagnosis was also based on the 3rd edition of the World Health Organization - Classification of Hematolymphoid Tumors.

Requests for inclusions in the text of the article were answered.

3) The matching criteria should also described. On page 6, row 137, the authors wrote “The studied cases were paired by sex and age group with 455 healthy individuals (controls)”. First, the correct term would be “approximately 2:1 matched”. Second, within what age range was matching acceptable? Were they 5-year or 10-year age-groups? From Table 1, seems that it was 20–year, which would be quite large. However, based on the small difference in the mean age between cases and controls reported on page 9, rows 212-214, the matching was effective in making cases and controls comparable by age. So, please, explain in the discussion that the broad age categorization did not affect the comparability of cases and controls by gender and case-control status. 

Answer: Cases and controls were matched for age and sex. In this study, the matching age range acceptable was plus or minus 10 years (25 -34; 35 - 44, 45 - 54; 55 - 64; 65 - 75). Due to the better use of the statistical power applied to multiple comparisons combined with the cutoff points pre-established in the literature, the categories of “20-39”, “40-59” and “> 60 years” were considered in the analyses. To verify that the categorization did not affect the comparability of cases and controls, note that the difference observed between groups was not significant. 

4) Third, on rows # 141-142, the authors wrote that “...for each new case diagnosed and included in the study, a new control was chosen...”. Well, the controls were twice as the cases, so at least two controls has to be matched to each case. Fourth, the 455 controls were presumably “healthy” not “all healthy”, as only those with a history of cancer were excluded.

Answer: The reviewer's requests were considered in the text.

5) The source of the data is unclear. The rows 167-181 on page 7 describe a list of data that were “collected. A questionnaire and interviews are mentioned but was it a structured interview with a standardized questionnaire? Was the questionnaire administered in person by trained nurses/interviewers or self-administered? Was it a consent form signed before the interview?

Answer: The individual interviews of the research participants were carried out by qualified and trained professionals from the Environment, Work and Cancer Technical Area of the National Cancer Institute (INCA). Data collection was performed after signing the Free and Informed Consent Form (TCLE). A standardized and pre-tested questionnaire was used with questions aimed at obtaining sociodemographic, clinical and occupational information.

On the day of the medical consultation or some routine procedure at the hospital, the patients were invited to participate in the research by professionals from the INCA research group and those who agreed to participate in the research signed the Free and Informed Consent Form (TCLE).

In the same period, depending on who entered the hospital, companions and visitors of patients diagnosed with non-hematological neoplasms undergoing medical care at the outpatient clinics or wards of Cancer Hospital I (HCI) of the National Cancer Institute INCA were invited to participate in the survey to compose the group control".

In addition, there was a need to recruit donors of biological materials at the Hemotherapy Service (HC1/INCA), because most of the companions and visitors of patients diagnosed with non-hematological neoplasms undergoing medical care at the outpatient clinics or wards of the HCI hospital in INCA were female. Therefore, on recruitment days, donors of biological materials at the Hemotherapy Service (HC1/INCA) were also invited to participate in the research.

The Informed Consent Term was signed before the interview and the questionnaire was applied by a health professional from INCA.

6) Please note that, on page 6 row 150, the number of participating controls is 577, while it is 557 in the flow diagram in figure 1. In the diagram and in the text the refusals to participation are not indicated. While aware of the higher participation rate of hospital controls vs population controls, this reviewer does not believe that every one of the eligible controls accepted. The 20 missing in the count (3.5%) are proportionally identical to the refusals among the cases (which, by the way, is 3.6 not 3.5%). 

Answer: 557 individuals were invited to participate in the control group of the study. However, 102 (18.9%) did not meet the inclusion criteria or refused to participate in the study, as shown in the diagram (Figure 1). As the individuals in the control group were visitors to the wards of patients with non-hematological cancers, the possibility of selection bias was reduced.

7) The occupational information was limited to the main occupation in the last 20 years lasting one year or more. As this article mainly focuses on occupational risk factors, this reviewer would have expected a more detailed work history. Those unaware of the Brazilian labour market situation, as this reviewer is, cannot judge whether a substantial part of short-term, low-paying and possibly highly exposed jobs were excluded from study.

Answer: In the occupational history, all occupations with working time greater than 1 year were considered. The research group chose to establish a cut-off point greater than 1 year for exposure time due to the difficulty in making a causal link between short exposure time and the development of non-Hodgkin's lymphoma. The questionnaire contained closed questions about occupational exposures that were self-reported.

The vast majority of our participants were over 40 years of age. In Brazil, job turnover 20-30 years ago was less common than today. People valued staying at work, stability. Thus, it is believed that there would be very few reports of less than 1 year of work in our study.

In Brazil, the best-known exposures related to the development of lymphomas are the use of pesticides in agriculture and benzene at fuel retail stations, mechanic workshops or refineries. In Brazil, the predominant agricultural model is still family farming. The rural worker, normally, starts his working life in agriculture and remains in it in his youth and in his adult life. Those exposed to benzene mostly work in large refining or fuel distribution companies, in the formal market, which reduces the common turnover observed in informal jobs.

8) Also, who did assess the exposures? Where these self-reported by the study participants? Was an expert occupational physician/industrial hygienist involved in the exposure assessment? Were jobs and industry coded in a standard format? Was a job-exposure matrix applied to the occupation and industry codes of study subjects? Apparently not, as the generic definitions of solvents, paints, metals, pesticides, cleaning products, biological material, animal and viscera material, industrial radiation and solar radiation seem to suggest. The current standards of occupational epidemiology research are far ahead in respect to this stage. Please, describe this as a limitation in interpreting the findings from this study. 

Answer: The exposures were evaluated by professionals (epidemiologists and toxicologists) who work in the Technical Area Environment, Work and Cancer of the National Cancer Institute.

Data related to occupations, economic activities and exposure of cases and controls were collected through a structured individual questionnaire, which contained a detailed occupational history recall. The participant responded about which agents he was exposed to and for how long, in the last 20 years, for each reported occupation.

Reports of exposures were recorded from the questionnaire that retrieved the occupational history of each participant, through closed questions, such as: “During occupation X, were you exposed to benzene? (Yes/No)". And so it went on for all the other exhibitions.

The authors chose to present the results by lymphoma subtype and keep the exposures grouped (solvents, hydrocarbons, pesticides, metals, visceral and animal material, and solar radiation), without specifying which solvents were reported, which metals or pesticides, etc. 

Afterwards, specialist professionals from the Environment, Work and Cancer Technical Area, trained in the construction of occupational exposure matrices, which are part of the Carex-Brazil Project (carcinogen exposure-Brazil), worked on these exposure variables that were disaggregated and grouped into groups alike. The description of the exposures reported by the study participants, which were contained in each group, can be seen at the bottom of Table 3 and in the following text:

Any solvents: turpentine, tar, asphalt, benzene, benzine, cleaners or degreaser, synthetic dye hardener, formaldehyde, gasoline, used motor oil, diesel oil, lubricating oil, crude oil, tar, kerosene, reducing agent (not specific), remover (not specs), thinner.

Hydrocarbon solvents: tar, benzene, gasoline, kerosene, thinner (not spec), remover (not spec), thinner. 

Metals: chrome, iron, and lead

Pesticides: Pesticides, insecticide, rodenticide, ant killer

This information and explanations were placed in the Methods part of the article.

9) Besides, a little background of what occupations contributed to the exposure to solvents and metals might help the interpretation. Likewise, the prevalent type of crops and some information on the best selling pesticides (insecticides, herbicides and fungicides mainly) in the study area, and not just overall Brazil, might also be helpful.

Answer: The database analyzed so far has the results of the history of occupations within economic activities, that is, by large groups of economic activities, such as: (a) for solvents = extractive industries, civil construction, water, sewage , waste management and decontamination activities; (b) metals = water, sewage, waste management and decontamination activities, civil construction, manufacturing industry and extractive industry (CNAE: national classification of economic activity, 2017; Brazilian institute of geography and statistics. available at: http:/ /www.cnae.ibge.gov.br/.). The study participant informed/reported his occupation and the project researchers classified the economic activity based on the National Classification of Economic Activity (CNAE).

The most predominant types of crops in the state of Rio de Janeiro are vegetables, legumes, tubers and citrus fruits. The most used and sold pesticides in the study area in the year 2017 were glyphosate, paraquat, deltamethrin, picloram, malathion, imidacloprid, lambda-cyhalothrin etc.

10) On page 7, rows 198-199, the authors wrote: “By unconditional logistic regression, multivariate models were estimated to analyze exposure...” This is an awkward sentence. The authors refer to the use of unconditional logistic regression to estimate the Odds Ratio for specific disease entities associated with the exposures under scrutiny. However, the miracle of multivariate analysis is to adjust the risk estimate of the covariate of interest by the concurrent effect of other covariates. Usually age, gender, and education are included by default. Other covariates might be relevant, such as income, marital status, and residence whether in metropolitan Rio de Janeiro or elsewhere. So, please, list what covariates were included in the logistic model and whether a forward or backwards stepwise procedure was adopted to select them. It is the personal opinion of this reviewer that, in the omics era, race categories based on the colour of the skin do not make any sense and are not surrogate of anything. Therefore, he would suggest to omit it from Table 1 , from page 9, rows 214-215, and from page 8 rows 304-311, or to justify the use of this otherwise useless category.

Answer: The covariates used in the study were: gender, age, skin color, family income, place of residence, level of education, smoking, alcoholism, past illnesses, and also family history of hematological cancer, including history of NHL (described in the “Data collection” section and in the footer of table 3). To select the covariates that composed the adjustment to the multivariate models, bivariate analyzes of the covariates with the study outcome (NHL) were initially performed, considering a p-value < 0.250 (Hosmer and Lemeshow, 2000). For multivariate analysis, logistic regression models were used using the manual stepwise forward methodology, where the occupational exposure of interest was tested and the adjustment of each model was evaluated by entry each of the previously selected covariates.

The general adjustment of the models was performed by testing the hypothesis that the initial model would present an adjustment to the data with the likelihood function, and that it reflects the probability that the estimated model represents the analyzed data set. For this test, the likelihood (-2LL), a chi-square distribution (n –p) corresponding to the degrees of freedom is assumed, in which 'n' represents the number of respondents and 'q' the number of model parameters.

Wald statistics and the change in the OR of the exposure of interest when including the potential confounder in the model were also considered. Afterwards, those variables with the best result in the Wald statistic, which caused the most modification in the OR of interest and which reduced the -2LL of the model, were selected for the final model.

The text of the methodology referring to this session was modified.

The variable skin color/race was removed as suggested by the reviewer.

11) Table 2 indicates that three controls and one case had a diagnosis of myelodysplasia. Myelodysplastic syndromes were so defined for the first time by the French-American-British Working Group in 1976, and considered as a precursor stage for acute myeloid leukaemia. This seems odd, as the authors declared to have excluded subjects with any form of cancer or visiting parents affected by haematological diseases from the control group. Please, justify the inclusion of the myelodysplasia cases or rerun the analysis after excluding them.

Answer: Statistical analyzes were performed again excluding the 3 controls and 1 case of myelodysplasia.

12) Cases of follicular lymphoma were 40 and those of T-cell lymphoma were 26. Why Table 3 shows results for the second and DLBCL, but not for follicular lymphoma? Because of the recent observation of the elevated risk of follicular associated with use of glyphosate, and the note in the discussion about glyphosate being one of the most used pesticides in Brazil, perhaps and analysis of the 40 cases of follicular lymphoma in relation to pesticide use might be interesting.

Answer: The requested analyzes were carried out and included in table 3.

13) When presenting the results, the authors gave too much emphasis to the positive associations with risk of T cell lymphoma, which were generated by small numbers. An introductory sentence such as “Risk estimates were stronger for T-cell lymphomas, but the wide confidence intervals suggest caution in interpreting them.” In particular, the 5-fold excess risk of T-cell lymphomas associated with exposure to metals suffers from three drawbacks: first the generic definition of the risk factors, second the extremely wide confidence interval, and third the inverse dose-response trend with years of exposure. Likewise, the almost 17-fold excess risk of T-cell lymphoma associated with generic exposure to pesticides seems to be mainly due to the wild ups and downs in the risk estimates generated by the small number of the exposed for 10 years or more. 

Answer: The results were reviewed with insertion of the introductory phrase suggested by the reviewer. In the conclusion, a sentence was also inserted that reduces the weight initially given to results with positive associations of broad magnitude, but with wide confidence intervals. With the review of the results, the magnitudes of the measures also changed and some were reduced.

Regarding the inverse trend of dose-response with years of exposure, we did not observe this. Only a reduction in the magnitude of exposure was observed over periods greater than 10 years for some subtypes and exposures, but the OR was still >1 and, in several crosses results, were statistically significant.

14) As it concerns the sanitizing products, this reviewer is puzzled by not finding a T-cell lymphoma risk estimate in Table 3, associated with the and 7 cases exposed for less or more than 10 years.

Answer: In the first analysis performed, the bivariate analysis, no association was found between exposure to sanitizing agents and T-cell lymphoma (p< 0.250), therefore, multivariate analyzes were not performed. This methodology was applied for all exposures and histological types of NHL. The following information has been included in the footnote to Table 3:

“Exposure models were not estimated for bivariate analyzes with p>0.25.”

15) This reviewer also suspects that the association with exposure to solar radiation was mostly, if not entirely, driven by farmers. Were there any non-farmers exposed to solar radiation (fishermen for instance or any other open-air occupations)?

Answer: The percentage of controls shows a difference in frequency distribution when compared to the percentage of non-Hodgkin's lymphoma cases in general. 

It can be observed in the first subcolumn (“Controls/Cases”) of each subtype in Table 3 that as the exposure time increases, the cases of lymphoma for all subtypes increase (example below), except for the T-cell subtype.

 Solar radiation Control Case (Total NHL) Total

 N % N % N %

No exposure 353 72,2 136 27,8 489 100,0

1 to 10 years 56 60,9 36 39,1 92 100,0

11 to 20 years 21 44,7 26 55,3 47 100,0

Total 430 68,5 198 31,5 628 100,0

16) The first part of the discussion illustrates the results of lifestyle (smoking and alcohol), and clinical variables (diabetes, viral infections) that were generated by univariate n-by-two tables and not by the multivariate analysis. There are also several drawbacks in the way these were classified; for instance the cut-off for being classified as an alcohol drinker was ridiculous (one drink per week). Considering the U-shaped curve of its relationship of alcohol-related cancer and other diseases, perhaps a categorization of drinkers as occasional and regular would have been more meaningful. Besides, there is no distinction between type 1 and Type 2 diabetes, which is mostly a feature of the metabolic syndrome. 

Answer: A category present in the questionnaire regarding the number of weekly doses of alcoholic beverage per study participant was included in the table.

As for the distinction between type 1 and type 2 diabetes, unfortunately we do not have this data. It is a limitation of the study.

17) Viral diseases, including HTLV infection, were apparently self-reported (there is no mention of serological testing). However, while there is no discussion about the more numerous self-reported cases of HCV, a disproportionately long paragraph is dedicated to discuss the HTLV finding. Both would be worth a short mention with proper caution because of the small study size, the small number of positive answers, and the crude testing of the difference. 

Answer: Request granted and the text has been rewritten. The patients' HTLV cases were checked from the medical records, but the limitation of the study is that the controls did not undergo serological tests.

18) A long discussion on benzene follows without explaining whether benzene and the other aromatic hydrocarbons were part of the category of hydrocarbon solvents. In fact, such a definition would include also the widely used aliphatic hydrocarbons (such as n-hexane) and the less refined petroleum-derived solvents, such as mineral spirit, gasoline, and naphta. Not to mention the weak finding on metals and T-cell lymphoma or that on solar radiation. 

Answer: Benzene was one of the most reported hydrocarbons by study participants within the solvent category group. There have also been reports of benzene, formaldehyde, gasoline, motor oil, lubricating oil, kerosene, etc. We agree that the discussion cannot have a broad focus only on occupational exposure to benzene, since most workers who are exposed to benzene are also exposed to other agents such as xylene, toluene, ethylbenzene (present in gasoline), different types of oils, greases and other solvents.

It should be noted that in the HPA group there is also benzo(a)pyrene (reported by the participants as pitch, asphalt, etc.), used in the production of coke, in the exhaust of diesel engines, gasoline engines, etc., as well as in untreated oils.

Any solvents: turpentine, tar, asphalt, benzene, benzine, cleaners or degreaser, synthetic dye hardener, formaldehyde, gasoline, used motor oil, diesel oil, lubricating oil, crude oil, tar, kerosene, reducing agent (not specific), remover (not specs), thinner.

Hydrocarbon solvents: tar, benzene, gasoline, kerosene, thinner (not spec), remover (not spec), thinner.

The discussion text has been rewritten.

19) The whole discussion needs to be reshaped and shortened. The limitations (small study size, occupational history limited to the longest most recent job, reliance on self reported exposures, generic definition of the relevant occupational risk factor, multiple comparisons) need to be properly highlighted and discussed.

Answer: Some specific excerpts from the discussion requested by reviewers have been rewritten and redrafted. If it is still necessary, we can make new changes and accept new suggestions from reviewers.

20) The conclusion, instead, overemphasize the observed positive associations.

Answer: The conclusion has been restated in order not to overemphasize the observed positive associations.

Minor points

Introduction, page 3, rows 81-83: “NHL is among the top ten most common types of cancer in Brazil, each year, for the triennium 2020-2022. An estimated 6,580 and 5,450 new cases occur annually in Brazilian men and women”.

The triennium 2020-2022 is not over yet. So, the authors might rephrase this sentence. This reviewer suggests something like “...common types of cancer in Brazil, each year. For the triennium 2020-2022, an estimated 6,580 and 5,450 new cases will occur”.

Answer: Request answered

Introduction, page 4, row 84: this is a “rate” not a “risk”. Please, use the correct term.

Answer: Request answered

Introduction, page 4, rows 97-100: “Other occupational factors related to NHL include solar radiation (12); ionizing radiation (11); organic dust (grain, fabric, and wood dusts); and mineral and metal dust; however, evidence of the association of these factors with NHL is not yet consistent (7).”

Well, these factors have been investigated, but the evidence for their association with an increase in risk is not simply “not consistent”: it does not exist. For instance, the reference # 12 found an inverse association between exposure to solar radiation and NHL risk. Note that reference # 11 refers to agricultural work not ionizing radiation; a range of suitable citations or a review (for instance Schmitz-Feuerhake I et al. Ann Hematol. 2022;101(2):243-250) might better serve the purpose. Also, a citation might accompany the mention of organic dust (Cocco P et al. Scand J Work Environ Health 2021;47(1):42-51) and mineral and metal dust (Fritschi L et al. Cancer Causes Control. 2005;16(5):599-607).

Answer: Regarding reference number 12, we understand that the authors found a positive association between NHL and occupational exposure to solar radiation, but not between the subtypes. Here's a summary:

“Occupational sun exposure was positively associated with the risk of NHL 1.14 (95% confidence intervals: 1.05, 1.23; I2=25.4% p for heterogeneity =0.202) in Caucasian population. Common subtypes of non-Hodgkin lymphoma and ultraviolet exposure had negative results. The pooled odds ratios was 1.16, (95%confidence intervals: 0.90, 1.50) for T-cell non-Hodgkin lymphoma; 0.79, (95%confidence intervals: 0.61, 1.02) for B-cell non-Hodgkin lymphoma; 1.13, (95%confidence intervals: 0.96, 1.34) for chronic lymphocytic leukemia; 1.25, (95%confidence intervals: 0.95, 1.64) for males; 1.49, (95%confidence intervals: 0.99, 2.25) for females.”

Regarding ionizing radiation, the Schmitz-Feuerhake results can be seen at: [https://www.ncbi.nlm.nih.gov/pmc/articles/PMC8742808/table/Tab3/?report=objectonly] where it presents a series of studies with RR >1 considering occupational exposure and ionizing radiation in different occupations.

With regard to dust, a citation of the work by Cocco and Fritchi was inserted, as suggested.

Methods, page 5, rows 118-119. The comparison with Portugal is unnecessary. Please, delete this sentence.

Answer: Sentence deleted

Methods, page 5, row 129. This reviewer is unaware of the need of establishing the sampling error in calculating the statistical power of a case control study. Please make a reference or omit mentioning the sampling error.

Answer: Request answered

Methods, page 5, row 132. “OR = 2,5”. Please, use the period to separate the decimal.

Answer: Request answered

Page 6, methods, rows 144-154. Please move these two paragraphs immediately after the statistical power paragraph and before indicating the final number of cases included in this study. Please, replace the terms “That way...” with “Therefore...”.

Answer: Request answered

Page 6, row 144: the verbal form “were” should be included after “...for [the] cases...”

Answer: Request answered

Page 7, row 174. What is the point of mentioning the minimum salaries? Wouldn’t the family income be enough? How was the amount of 5BRL selected for the cut-off? Does it make any sense showing also the mean income in Table 1?

Answer: In Brazil, the variable "number of minimum salaries" is used to classify individuals into the categories: poor, middle and upper class. There is no cut-off in the amount of 5 Brazilian NMW (National Minimum Wage). This is the number referring to 5 minimum wage. In order not to cause confusion, this value was replaced by the average family income and we put this information in table 1.

Page 7, row 176. “cellular use” perhaps refers to cellular phone. Please, rephrase.

Answer: Request answered

Page 8, row 195. “Missing values were excluded from the analysis”. Table 1 should indicate how many missing values were there for each variable of interest.

Answer: The “no information”/missing values information was placed in tables 1 and 2.

Page 8, Results, row 209. “...and T-cell lymphoma (30; 14.0%)”. As it is written seems the authors are saying that T-cell lymphomas were also prevalent. The sentence should be written as it follows “...and T-cell lymphomas were 30 (14.0%).”

Answer: Request answered

Page 11, rows 234-235: “We also found a difference in the use of cell and wireless phones (p > 0.001)” and Table 2. The sentence is formulated ambiguously. In fact, the point is that controls used cell phones and wireless phones more than the cases, which might be a consequence of the lower income and lower education of the cases in respect to the controls. Therefore, the phrasing should be explicit about the fact that “use of cell phones and wireless phones was more frequent and more prolonged among the controls than the cases”.

Answer: Request answered

Page 13, row 254, same page, row 272: use the terms “years of exposure” instead of “exposure time”.

Answer: Request answered

Page 18, row 303: delete the phrase “by histological subtype, and these data are”

Answer: Request answered

Page 18, rows 304-311. Delete the whole sentences.

Answer: Request answered

Page 18, rows 314-315: “Among the NHL cases, some patients resided in other regions of Rio de Janeiro.” Unclear statement. Does it mean that a proportion of the cases larger than controls resided in regions other than metropolitan RJ?

Answer: Around 80% of the cases resided in the metropolitan region of the state of Rio de Janeiro and part the same populational base. It is important to note that the other municipalities in Rio de Janeiro are very close to the metropolitan area, since the state of Rio de Janeiro is a small state in terms of geographic area. This variable, place of residence, was controlled in the multivariate analysis.

Page 19, row 319. Delete the colloquial term “your” and replace it with the article “the”.

Answer: Request answered

Page 19, rows 321-327. Smoking is not a strong risk factor for lymphoma, apart from a weak association with risk of follicular lymphoma. Besides, the case-control difference in the proportion of smokers was not the result of the multivariate analysis but of a simple 2x2 table, and current smokers prevailed among the controls. This reviewer suggest the authors to stick to their findings and avoid discussing meaningless findings.

Answer: The initial objective of the study was to evaluate occupational variables and lymphomas. We may have overestimated the association between the NHL and smoking, because when we look at the “smoke load”, we see that cases smoke more than controls.

Please, format the list of references in a consistent way following the journal guidelines.

Answer: The list of references was formatted following the journal's guidelines.

Unfortunately, due to the Christmas break and the end of the year, as well as the summer vacation in Brazil, we had little time to answer reviewers' questions. Therefore, we are available to answer questions that have not been clarified. 

Thanks again for the opportunity and the reviewers' review!

---

## [Decision Letter · Decision Letter 1]

18 Apr 2023

PONE-D-22-29162R1INVESTIGATION OF OCCUPATIONAL RISK FACTORS FOR THE DEVELOPMENT OF NON-HODGKIN LYMPHOMA IN ADULTS: A HOSPITAL-BASED CASE-CONTROL STUDYPLOS ONE

Dear Dr. Sarpa,

Thank you for submitting your manuscript to PLOS ONE. After careful consideration, we feel that it has merit but does not fully meet PLOS ONE’s publication criteria as it currently stands. Therefore, we invite you to submit a revised version of the manuscript that addresses the points raised during the review process.

When revising your manuscript, please remember to properly address the given comments, including proper analysis of the obtained data, thorough discussion of results, and proper formatting of the text as per PLOS ONE guides.

We look forward to receiving your revised manuscript.

Kind regards,

Elingarami Sauli, PhD

Academic Editor

PLOS ONE

Reviewers' comments:

Reviewer's Responses to Questions

**Comments to the Author**

1. If the authors have adequately addressed your comments raised in a previous round of review and you feel that this manuscript is now acceptable for publication, you may indicate that here to bypass the “Comments to the Author” section, enter your conflict of interest statement in the “Confidential to Editor” section, and submit your "Accept" recommendation.

Reviewer #2: (No Response)

2. Is the manuscript technically sound, and do the data support the conclusions?

Reviewer #2: No

3. Has the statistical analysis been performed appropriately and rigorously? 

Reviewer #2: No

4. Have the authors made all data underlying the findings in their manuscript fully available?

Reviewer #2: Yes

5. Is the manuscript presented in an intelligible fashion and written in standard English?

Reviewer #2: No

6. Review Comments to the Author

Reviewer #2: This reviewer appreciates the authors’ effort to revise the previous version. However, this new version still requires some additional effort. The first and major point is the inclusion of blood donors among the controls to balance the gender structure with the cases. Blood donors, by definition, are healthier than the general population and, therefore, their inclusion makes any attempt of exploring the association with the health history severely and irreparably biased. Nonetheless, the paper has some good points too, which can be saved provided that the authors keep complying with the reviewers’ comments and suggestions.

One suggestion would be to omit the analysis of the past health history and focus on the analysis of occupational risk factors. The adjustments by alcohol and smoking can at least partially fix the selection bias related to the blood donors’ healthier status. If the authors wish to maintain the analyses on past illnesses, they should limit them to the 309 controls recruited among the patients’ companions and visitors.

In any case, the selection bias introduced by using blood donors as controls needs to be openly acknowledged and discussed among the limitations of this study.

The discussion should start with the main results, which are those related to the occupational risk factors. Lifestyle variables were only evaluated in the univariate analysis and smoking, and alcohol only considered as covariates in the regression models. Therefore, a short comment would be enough.

Consistently with the bias generated by using blood donors as controls, all the results and discussion on diabetes, HTLV and HCV should be omitted. Those on diabetes and perhaps HCV might be briefly commented upon in the case the authors repeated the analysis after excluding the 143 blood donors. Also, rerun the logistic regressions that included diabetes and HHV8 infection as covariates. The results on metals seem also contradictory; a short comparison with the literature findings would be enough.

From Table 2, it seems that the cases had a lower socio-economic status, but the list of covariates in the footnotes of table 3 do not include any specific adjustment. This might have contributed to generate some possibly spurious excess, such as that with exposure to sunlight. This reviewer suggests including education in the regression models, as listed on line 193 under the subheading on statistical methods as a surrogate for socio-economic status. Also, include occupation as a farmer as a covariate in the logistic regression predicting risk of NHL & subtypes associated with exposure to sunlight.

At the end of the discussion, the authors should insert a sub-section on the study limitations: selection bias, multiple comparisons, recall bias due to the self-report of the occupational exposure, years of exposure as a poor surrogate of cumulative exposure, lack of exposure assessment by experts to define other exposure metrics, such as intensity and frequency of exposure, the generic definition of the exposure categories, possible confounders unaccounted for (socioeconomic status, agricultural work and exposure to sunlight). The effects of all the above on the risk estimates needs a thorough discussion.

The text requires an in-depth revision of the style, grammar, and typos. This reviewer went through this type of editing as well for the abstract, introduction, methods and results, but the list in the attached document, though long, might not be exhaustive.

7. PLOS authors have the option to publish the peer review history of their article (what does this mean?). If published, this will include your full peer review and any attached files.

Reviewer #2: No

---

## [Author Response · Author response to Decision Letter 1]

28 Jul 2023

Reply to reviewer – PLOS ONE – Second round - April 2023 – June 2023

This reviewer appreciates the authors’ effort to revise the previous version. However, this new version still requires some additional effort. The first and major point is the inclusion of blood donors among the controls to balance the gender structure with the cases. Blood donors, by definition, are healthier than the general population and, therefore, their inclusion makes any attempt of exploring the association with the health history severely and irreparably biased. Nonetheless, the paper has some good points too, which can be saved provided that the authors keep complying with the reviewers’ comments and suggestions.

Answer:

The controls were selected from the companions and visitors of patients diagnosed with solid cancers and also from the waiting room of the Hospital's blood bank, which also receives mostly visitors and patients' companions. This procedure was adopted to minimize the effect of a possible control selection bias. In this way, the controls selected at the Blood Bank did not differ from the controls collected in the Hospital wards, as they are also visitors or companions of the patients. It should also be noted that INCA is not a blood center. In order to not cause problems in the interpretation, the term “biological material donors” was replaced by “blood bank controls”.

One suggestion would be to omit the analysis of the past health history and focus on the analysis of occupational risk factors. The adjustments by alcohol and smoking can at least partially fix the selection bias related to the blood donors’ healthier status. If the authors wish to maintain the analyses on past illnesses, they should limit them to the 309 controls recruited among the patients’ companions and visitors. 

In any case, the selection bias introduced by using blood donors as controls needs to be openly acknowledged and discussed among the limitations of this study. 

The discussion should start with the main results, which are those related to the occupational risk factors. Lifestyle variables were only evaluated in the univariate analysis and smoking, and alcohol only considered as covariates in the regression models. Therefore, a short comment would be enough. 

Ok.

Consistently with the bias generated by using blood donors as controls, all the results and discussion on diabetes, HTLV and HCV should be omitted. Those on diabetes and perhaps HCV might be briefly commented upon in the case the authors repeated the analysis after excluding the 143 blood donors. Also, rerun the logistic regressions that included diabetes and HHV8 infection as covariates. 

Parts of the diabetes, HTLV, and HCV discussions were omitted as directed.

The results on metals seem also contradictory; a short comparison with the literature findings would be enough. 

Ok

From Table 2, it seems that the cases had a lower socio-economic status, but the list of covariates in the footnotes of table 3 do not include any specific adjustment. This might have contributed to generate some possibly spurious excess, such as that with exposure to sunlight. This reviewer suggests including education in the regression models, as listed on line 193 under the subheading on statistical methods as a surrogate for socio-economic status. Also, include occupation as a farmer as a covariate in the logistic regression predicting risk of NHL & subtypes associated with exposure to sunlight

All potential confounding factors of the theoretical model were tested, as well as the level of education was used in the first part of the analysis with the selection criterion (Wald p-value < 0.025). In the multivariate analysis, other socioeconomic variables quantified the variability of response variables, such as place of residence and family income. Only three cases self-declared one of the occupations as a farmer and two as cattle ranchers, therefore, it is not possible to carry out this specific analysis.

At the end of the discussion, the authors should insert a sub-section on the study limitations: selection bias, multiple comparisons, recall bias due to the self-report of the occupational exposure, years of exposure as a poor surrogate of cumulative exposure, lack of exposure assessment by experts to define other exposure metrics, such as intensity and frequency of exposure, the generic definition of the exposure categories, possible confounders unaccounted for (socioeconomic status, agricultural work and exposure to sunlight). The effects of all the above on the risk estimates needs a thorough discussion.

Done.

The text requires an in-depth revision of the style, grammar, and typos. This reviewer went through this type of editing as well for the abstract, introduction, methods and results, but the list in the attached document, though long, might not be exhaustive.

The final version of the article after all the revisions suggested by the reviewers will be sent to the translator.  

PLOS One 

pone-d-22-29162r1

Investigation of occupational risk factors for the development of non-Hodgkin lymphoma in adults: a hospital-based case-control study.

Additional list of editings.

Abstract

The abstract is too long, includes unnecessary information, and omits important results. For instance, the long sentence in lines 38-45 can be replaced with “We explored the association of NHL risk occupational and lifestyle exposures [and health history, if the authors wish to pursue their analysis using only companion or visitor controls] with a case-control study design in 214 adult patients and 452 population controls in Brazil.” 

Ok. 

Remove the sentence about ethical approval in lines 47-49 and paste it at the end of the Methods section. 

Ok. Removed

The abstract should report the odds ratio and 95% confidence interval associated with ever exposure to the occupational exposures showing a significant association, and the respective p-value of the test for trend. The p-value of the test for trend should also be reported in Table 3. The lines 49-56 in the abstract should be replaced with the following: “risk of NHL (any subtype), B-cell lymphoma, and T-cell lymphoma was elevated among the ever exposed to solvents [give the Odds ratio and 95% confidence interval associated with ever exposure (=all the exposed)], pesticides [Odds ratio and 95% CI for the ever exposed], meat and meat products [instead of animal and visceral materials] [Odds ratio and 95% CI for the ever exposed], and sunlight [instead of solar radiation] [Odds ratio and 95% CI for ever exposed]. A significant upward trend with years of exposure was detected for solvents and hydrocarbon solvents (NHL (any subtype) p-value for trend), B-cell lymphoma (p for trend), and T-cell lymphoma (p for trend)), pesticides (NHL (any subtype), p for trend) , B-cell lymphoma (p for trend), DLBCL (p for trend), and T-cell lymphoma (p for trend), raw meat (NHL (any subtype) (p for trend) and DLBCL (p for trend), and sunlight (B-cell lymphoma (p for trend), DLBCL [? Not sure] (p for trend), and follicular lymphoma [if still significant after adjustment for agricultural work] (p for trend)”. 

Ok. It was done.

The results for metals are not worth reporting in the abstract as they are non-significant and show an inverse trend with years of exposure.

Ok.

Drop the sentence in lines 57-58 from “…can be extrapolated to other countries” through “…Brazil” and, in the following sentence, on line 58, delete “…will”.

Ok

Introduction

On line 67, replace “which” with “and”.

Ok

Lines 71-72. In all case series this reviewer is aware of, CLL/SLL is among the most frequent mature B-cell subtypes, sometimes the most frequent. It might be less so in South America,or Asia, but certainly is not definable as a “less common” subtype”. A lesser frequency might be due to underdiagnosis because of the silent clinical course of most cases. Just omit mentioning CLL here.

Ok

Lines 96 – 98; include the two citations reported here with the name of the first author and year in the list of references, renumber the references progressively in order of citation, and replace name and year with the corresponding numbers throughout the manuscript.

Ok

Line 98: replace “not yet consistent” with “inconsistent”.

Ok

Line 99-101: Marant Micallef only summarized the IARC Working Groups’ conclusions, without making any further elaboration. So, this reviewer would suggest replacing the whole sentence with the following “The International Agency for Research on Cancer classifies 87 occupational exposures in groups 1 and 2A [13]”.

Ok

Line 102: replace “were attributed to non-Hodgkin’s lymphomas” with “were specifically associated with non-Hodgkin’s lymphoma”.

Ok

Methods

Line 113: replace “that reside” with “residing”.

Ok

Line 114; delete “…and were…”.

Ok

Line 117: replace “despite” with “although”.

Ok

Lines 118-121: delete from “…is a reference hospital…” through “…Government of Brazil,”

Ok

Line 123: delete the sentence “…and INCA is also responsible…”; insert the following after “…(CANCON)”: “INCA is a national institute, part of the Ministry of Health of the Federal Government of Brazil, responsible for providing care to around 7.000 new patients per year and…“.

Ok

Line 128: delete “…it was decided to calculate”.

Ok

Line 128: insert “…was determined using…” between “…sample size” and “…with the following”.

Ok

Line 129: insert “case-control” between “2:1” and “ratio”.

Ok

Line 130: delete the closing bracket.

Ok

Lines 130-131: replace “…for a statistical power of 80%, it will be necessary 215 cases and 430 controls” with “…215 cases and 430 controls would be required to achieve a statistical power of 80%”.

Ok

Lines 134-138: replace the sentence currently in lines 134-138 with the following: “Eligible controls were visitors or companions of patients at INCA/HCI, aged between 25 and 75 years and free from any type of cancer at the time of participating in the study”. 

Ok

Line 139: insert the following after “…557 controls”: “matched to cases by gender and age-group (± 10 years)”

Ok

Lines 141: delete “…previously established…”. Replace “Regarding controls, 105 subjects (18.9%)…” with “…One -hundred-five candidates as controls (18.9%)…”.

Ok

Line 142: In case the authors wish to pursue in their analysis of health history, they should explain here the reason for selecting blood donors and indicate their number (N = 143). This reviewer would suggest including a sentence such as: “As most visitors of patients diagnosed with non-haematological neoplasms at the hospital's outpatient clinics or wards were female, it was necessary to recruit male blood donors (N = 143) at the hospital Haemotherapy Service.” If the authors decide of omitting the analysis of health history, just delete the whole sentence.

Ok

Lines 149-151: report here the number of cases for each NHL subgroup.

Ok

Lines 152-159: delete the whole paragraph.

Ok

Lines 160-162: move the Ethics note to the end of the Methods section.

Ok

Line 167: delete “…individual”.

Ok

Lines 178-183. Delete these sentences [see note about line 142].

Ok

Lines 184-188: Delete the whole sentence as it is repetitive.

Ok

Line 189: delete “Thus, in order…”

Ok

Line 190: replace “…collected on the subject’s” with “…abstracted for analysis”.

Ok

Line 192: delete “race (white, non white)”. [following my previous review, race was excluded as a covariate]

Ok

Line 193: replace “time of residence” with “years at the current residence”.

Ok

Line 195: replace “not employed” with “unemployed”.

Ok

Line 196-197: Table 1 does not show what was the unit for average income. 

Ok

Line 198 and the below paragraph: See the comments on line 142. Also replace “In relation to habits and past illness were included..” with “Among the lifestyle variables, we considered the following:…”.

Ok

Lines 201-204: delete from “…past illnesses…” through the end of the sentence.

Ok

Line 205: replace “…their…” with “…the…”

Ok

Line 206: replace “working time greater than 1 year” with “lasting one year or more”.

Ok

Line 206-209: delete the unnecessary explanation for cutting short-term employments.

Ok

Line 212: “biological material, animal and viscera material”. Please, explain what the biological materials would be or whether the authors meant “infectious agents”. Also, the animal and viscera material, as this reviewer understand, would be the meat aerosol in abattoirs and butcheries. If so, the definition should be raw meat droplets and aerosols. Some information on what occupation were related to such exposures should be included in the Table 3 footnote.

The question asked through the questionnaire is very broad. You are asked about exposure to biological materials in the work environment (eg such as blood, urine or other exposures).

Line 213: replace “...collected…” with “…self-reported…”.

Ok

Lines 232-235: replace “By unconditional logistic regression, multivariate models were estimated to analyse exposure to chemical, physical, and biological agents as well as the risk of NHL according to the cases “Total NHL”, “B-cell lymphoma”, “DLBCL”, “Follicular lymphom, and “T-cell lymphoma”. The covariates used in the study were:” with the following: “Risk of NHL (any subtype), B-cell lymphoma, DLBCL, Follicular lymphoma, and T-cell lymphoma associated with exposure to chemical, physical, and biological agents was analysed by unconditional regression. The multivariate regression model included the following covariates:” 

Ok

Line 236: delete “past illnesses” and “also”.

Ok

Lines 238-245: Replace the whole sentence with the following: “To select the covariates to be included in the final regression model, we followed a stepwise forward procedure testing first the covariates with a p<0.25 association in the univariate analyses of the covariates with NHL (any subtype)[15].”

Ok

Drop diabetes and HHV8 infection from the covariates in the logistic regression model.

Parts of the diabetes, HTLV, and HCV discussions were omitted as directed.

Lines 245-247 “replace the whole sentence with the following: “The Odds Ratio and its 95% confidence interval were calculated as the measure of association; the null hypothesis was rejected if the 95% confidence interval did not include unity.”

Ok

Results

The data reported in the first sentence should be moved to the Methods: the number of participants by gender would fit in line 160, before the flow diagram in figure 1; the number of cases of each NHL subtype would fit in lines 234-235. No supplementary material is needed.

Ok

Line 257-259: delete this sentence.

Ok

Table 1. Replace “time of residence” with “years of residence”. Please show what was the unit for family income.

Ok

Lines 280-282: Delete the sentence on past illnesses and the corresponding results from table 2.

Excluded.

Lines 291-293: delete from “…of occupational…” through “…histological subtypes.” 

Ok

Line 294: insert ”any of” between “exposed to” and “those substances..”.

Ok

Line 297: Risk increased also for several other subtypes and other exposures. Please. report here the p-value for the tests for trend.

Trend p-value values are described in table 3.

Lines 298-310. Move these sentences from line 296, before the sentence on tests for trend, and report in the text the risks associated with ever exposure (i.e. independent on years of exposure).

Ok

Lines 317-321: move the result on T-cell lymphoma and pesticides before reporting about the tests for trend (line 315). Also, make it shorter as the small number of cases and controls suggests caution. Something like: “the strongest association was observed with risk of T-cell lymphoma, although the small number of subjects suggests caution in the interpretation”.

Ok

The same suggestion is valid in reporting the results for raw meat aerosols and exposure to sunlight.

Ok

Table 3: Insert risk for ever exposure (all the exposed one or more years independent on years of exposure) for each occupational risk factor. Calculate and report the p-value for each test for trend. It is unclear why some results are missing and others with smaller number and presumably more unstable estimates are shown instead (see for instance B-cell lymphoma and dyes, metals, biological agents, and ionizing [and not industrial] radiation or T-cell lymphoma and sanitizing products vs T-cell lymphoma and metals. In the footnote, insert what occupations were more frequently represented among the exposto biological materials and sunlight.

Occupations that most represent exposure to biological materials: Nurse, nursing technician.

Occupations that represent the most exposure to solar radiation: tradesman, housemaid, housewife, electrician, driver, mason, painter, military police, security, salesman.

---

## [Decision Letter · Decision Letter 2]

11 Sep 2023

PONE-D-22-29162R2INVESTIGATION OF OCCUPATIONAL RISK FACTORS FOR THE DEVELOPMENT OF NON-HODGKIN LYMPHOMA IN ADULTS: A HOSPITAL-BASED CASE-CONTROL STUDYPLOS ONE

Dear Dr. Sarpa,

Thank you for submitting your manuscript to PLOS ONE. After careful consideration, we feel that it has merit but does not fully meet PLOS ONE’s publication criteria as it currently stands. Therefore, we invite you to submit a revised version of the manuscript that addresses the points raised during the review process.

When responding to reviewer comments please make sure to include proper justification for choice of control, detailed/thorough discussion of results and also limitations of the study, without forgetting improvement of grammatical errors (which are many).

We look forward to receiving your revised manuscript.

Kind regards,

Elingarami Sauli, PhD

Academic Editor

PLOS ONE

Journal Requirements:

Reviewers' comments:

Reviewer's Responses to Questions

**Comments to the Author**

1. If the authors have adequately addressed your comments raised in a previous round of review and you feel that this manuscript is now acceptable for publication, you may indicate that here to bypass the “Comments to the Author” section, enter your conflict of interest statement in the “Confidential to Editor” section, and submit your "Accept" recommendation.

Reviewer #2: (No Response)

Reviewer #3: All comments have been addressed

Reviewer #4: All comments have been addressed

2. Is the manuscript technically sound, and do the data support the conclusions?

Reviewer #2: Partly

Reviewer #3: Yes

Reviewer #4: Partly

3. Has the statistical analysis been performed appropriately and rigorously? 

Reviewer #2: Yes

Reviewer #3: Yes

Reviewer #4: Yes

4. Have the authors made all data underlying the findings in their manuscript fully available?

Reviewer #2: No

Reviewer #3: Yes

Reviewer #4: Yes

5. Is the manuscript presented in an intelligible fashion and written in standard English?

Reviewer #2: No

Reviewer #3: Yes

Reviewer #4: No

6. Review Comments to the Author

Reviewer #2: This second revised version needs additional work, as it did not entirely match the response letter and the issues this reviewer raised to the two prior versions of this paper. However, as stated in reviewing the original version, this paper has merits despite the awkward writing, the unchecked citations, the recurrent repetitions, and the long discussion on points irrelevant to the subject of the paper. This reviewer understands the difficulty in writing scientific reports for an international audience, particularly for scholars from non-English-speaking countries and he hasn’t changed his mind. The corrections and amendments would be too many to list them. Therefore, the authors might check the notes in the attached pdf of their article, which includes all the corrections and amendments this reviewer could make hoping they will endorse them and prepare a newly revised version that he will gladly reconsider.

New analyses are not necessary; on page 9, row 193, after smoking, please describe the way alcohol intake was categorized. On page 10, row 228, please avoid using the term “alcoholism” and replace it with “alcohol intake as, fortunately, most alcohol drinkers get moderate amounts and never got drunk in their lifetime. Also, omit listing “skin colour” as a covariate: based on the footnote to Table 3, it was never included in the logistic models, and it is socially, scientifically, and methodologically unacceptable.

Renumber and double check all the references, including their format and their pertinence to what reported in the text (Ref. # 15 is cited twice: in neither instance it seems relevant to what reported in the text).

Reviewer #3: Authors satisfied deeply all comments of reviewers. No other comments are provided, well done! Good work.

Reviewer #4: General comments

The study is very interesting and important because provides data on NHL and occupational and lifestyle risk factors in Brazil. The paper presents a good balance among the different sections and provides important and original results.

But the study presents also some limits in particular the choice of controls (blood donors to balance gender, that could be halthier than the general popolation and be source of selection bias) and the lack of a good exposure assessment (by experts or using a job exposure matrix), using information on self-report occupational exposure .

The paper would benefit from some additional comments more in depth in the discussion section on limitations represented by using blood donors among the controls and the crude exposure assessment.

Furthermore for some sentences of the paper would important to add more recent references

Specific comments

Introduction section

I suggest to add a reference to support the sentence at rows 79-80 of the last version of the paper

Rows 86-87 : I suggest to add reference on the IARC evaluation

Row 92 : I suggest to add references

Row 95 : I suggest to add more references

Row 96 : the sunlight risk is controversial recent paper suggested also a protective role of UV radiation

Row 100: I suggest to report the number of occupational exposures considering the more recent IARC evaluations (now the last published monographs is the 132 volume)

Methods section

I suggest to the authors to move to the results section the number of the cases and controls recruited and to specify the participation rate.

Rows 113-115: delete the sentence on ethics committee that was reported also at rows 239-241

The authors should specify how information on occupational risk factors was requested, has information on exposure to occupational agents been requested for each working period, a list of the specific agents was presented to each subjects?

What type of information has been requested for sunlight exposure? The authors should specify better for understanding how exposure to sunlight are collected.

Results

I suggest to report a comment on the difference in the socio economic status.

I suggest to move the sentence in the rows 277-278 to the methods section

Rows 283 -285 I suggest to delete results for controls and report the reference table 3 in brackets

In pag. 23 and pag. 24 the sentence “The wide presence of heavy metals ….” is repeated twice with different references.

I suggest to the authors to add more recent paper on exposure to pesticides ( insecticides and herbicides) and risk of NHL and not only reference n.36 and the IARC Monographs on pesticides evaluations .

Discussion section

In the last version of the paper a section on limitations are presented, but it should be useful to report more in deep comments

Considering selection bias represented by controls group of blood donors a descriptive analysis on habits and past illness separately for the two control groups could be useful to do for better discuss this important point.

The authors should discuss more in deep the limitation represented by using self reported occupational exposure, especially when the use of each specific agent are requested to the subject and recall bias might be possible.

Some comments on the results on sunlight exposure and pesticide exposure are to be added, considering that outdoor workers (such as the farmers) may be exposed to pesticides but also to ultraviolet radiation and it is difficult to disaggregate the possible risk of these concomitan exposures.

7. PLOS authors have the option to publish the peer review history of their article (what does this mean?). If published, this will include your full peer review and any attached files.

Reviewer #2: **Yes: **Pierluigi Cocco

Reviewer #3: No

Reviewer #4: No

---

## [Author Response · Author response to Decision Letter 2]

27 Dec 2023

I would like to thank the review of the editors and reviewers. 

The suggested corrections were essential to improve the manuscript.

We've made all the suggestions. We are available to make new corrections, if they are suggested. 

Best regards.

---

## [Editor Report · Decision Letter 3]

28 Dec 2023

INVESTIGATION OF OCCUPATIONAL RISK FACTORS FOR THE DEVELOPMENT OF NON-HODGKIN LYMPHOMA IN ADULTS: A HOSPITAL-BASED CASE-CONTROL STUDY

PONE-D-22-29162R3

Dear Dr. Marcia Sarpa,

We’re pleased to inform you that your manuscript has been judged scientifically suitable for publication and will be formally accepted for publication once it meets all outstanding technical requirements.

Kind regards,

Elingarami Sauli, PhD

Academic Editor

PLOS ONE
---

## [Editor Report · Acceptance letter]

10 Feb 2024

PONE-D-22-29162R3 

PLOS ONE

Dear Dr. Sarpa, 

I'm pleased to inform you that your manuscript has been deemed suitable for publication in PLOS ONE. Congratulations! Your manuscript is now being handed over to our production team.

Kind regards, 

on behalf of

Dr. Elingarami Sauli 

Academic Editor

PLOS ONE